# TAPTRv3: Spatial and Temporal Context Foster Robust Tracking of Any Point in Long Video

**Jinyuan Qu**[1,3*] **Hongyang Li**[2,3*] **Shilong Liu**[1,3]
**Tianhe Ren**[3] **Zhaoyang Zeng**[3] **Lei Zhang**[3†]
[1]Tsinghua University [2]South China University of Technology
[3]International Digital Economy Academy (IDEA)

## Abstract

In this paper, built upon TAPTRv2, we present TAPTRv3. TAPTRv2 is a simple yet effective DETR-like point tracking framework that works fine in regular videos but tends to fail in long videos. TAPTRv3 improves TAPTRv2 by addressing its shortcomings in querying high-quality features from long videos, where the target tracking points normally undergo increasing variation over time. In TAPTRv3, we propose to utilize both spatial and temporal context to bring better feature querying along the spatial and temporal dimensions for more robust tracking in long videos. For better spatial feature querying, we identify that off-the-shelf attention mechanisms struggle with point-level tasks and present Context-aware Cross-Attention (CCA). CCA introduces spatial context into the attention mechanism to enhance the quality of attention scores when querying image features. For better temporal feature querying, we introduce Visibility-aware Long-Temporal Attention (VLTA), which conducts temporal attention over past frames while considering their corresponding visibilities. This effectively addresses the feature drifting problem in TAPTRv2 caused by its RNN-like long-term modeling. TAPTRv3 surpasses TAPTRv2 by a large margin on most of the challenging datasets and obtains state-of-the-art performance. Even when compared with methods trained on large-scale extra internal data, TAPTRv3 still demonstrates superiority.

## 1 Introduction

Localizing points across different frames in a video is a long-standing problem (Sand & Teller, 2008). Recently, with the growing demand for the trajectory and visibility information of arbitrary points in videos for various down-stream tasks, such as video editing (Huang et al., 2023), SLAM (Teufel et al., 2024), and manipulation (Vecerik et al., 2024), the Tracking Any Point (TAP) task has gradually regained attention (Harley et al., 2022; Doersch et al., 2023; Karaev et al., 2024b; Li et al., 2024c;b; Neoral et al., 2024; Xiao et al., 2024; Wang et al., 2023; Tumanyan et al., 2024).

To solve this problem, some methods try to construct a 4D field (Wang et al., 2023) and track points in the constructed 3D space while also achieving 3D scene perception. Though promising, such methods are normally not general and have inferior performance. By contrast, more methods attempt to solve the TAP task directly from a 2D perspective (Harley et al., 2022; Doersch et al., 2023; Karaev et al., 2024b;a; Li et al., 2024c;b; Tumanyan et al., 2024; Aydemir et al., 2025; Zholus et al., 2025). Some of these methods follow the traditional optical flow estimation pipeline, RAFT (Teed & Deng, 2020) more specifically, and highly rely on the dense cost-volume (Xu et al., 2017) to perform point tracking like sparse optical flow. Although such methods have achieved impressive performance, the computation of dense cost-volume is resource-consuming, especially when the number of points, the length of videos, or the video resolution increases.

Inspired by recent visual prompt-based detection methods (Li et al., 2024a; Jiang et al., 2025), TAPTR (Li et al., 2024c) proposes a DEtection-TRansformer (DETR)-like framework, which regards

---

[*]Equal contribution, random listing order. Work done during an internship at IDEA Research.
[†]Corresponding author.

each tracking point as a point query and addresses the TAP task from the perspective of point-level visual prompt detection. TAPTRv2 (Li et al., 2024b) further improves TAPTR by eliminating the requirement of dense cost-volume input, as it will contaminate the point query's content feature and introduce redundancy. To compensate for its localization role, TAPTRv2 proposes an attention-based position update (APU) operation that utilizes key-aware deformable attention (Li et al., 2023) to compare a query point with a set of key sampling points and find a better position. With this improvement, TAPTRv2 obtains both a simpler framework and a better performance.

However, we find that TAPTRv2 still struggles with long videos due to its shortage of feature querying in both spatial and temporal dimensions in long videos, in which the target tracking points normally undergo increasing variation over time. In the spatial dimension, TAPTRv2 introduces key-aware deformable attention (Li et al., 2023) to extract features and directly perform position update by comparing the feature similarity between a query point and a set of surrounding sampled key points. However, when migrating this module from an object-level detection task to a point-level tracking task, TAPTRv2 overlooks the fact that the point-level query and key features, obtained simply through bilinear interpolation, are too local. This makes the resulting attention weights susceptible to noise. This instability significantly affects point tracking, which requires the most fine-grained spatial understanding. In the temporal dimension, the RNN-like long-temporal modeling in TAPTRv2 often suffers from the drifting problem, as the feature of a tracking point may be gradually affected by ambiguous surrounding features and unknown occlusion over time. Moreover, there is a significant discrepancy in video length between the current training and testing sets. The training set consists of short videos with fixed 24 frames, while the testing videos vary from 50 to 1300 frames in length. Excessive feature updates in long videos during testing further exacerbate the feature drifting problem. In our study, we also observe the existence of scene cuts in many videos. While such videos are the result of artificial editing, they are quite prevalent in public datasets. For instance, in TAP-Vid-Kinetics (Doersch et al., 2022), which is one of the challenging test sets, approximately 27% of the videos contain scene cuts. The lack of global matching in TAPTRv2 makes it hard to reestablish tracking effectively when a scene cut occurs with sudden, large motions.

With these insights, we propose to enhance the feature querying ability of TAPTRv3 in both spatial and temporal dimensions. For *spatial feature querying*, inspired by the prior 4D cost volume-based optical flow (Teed & Deng, 2020; Ilg et al., 2017; Sun et al., 2018; Wang et al., 2020; Jiang et al., 2021; Xu et al., 2021) and point tracking methods (Cho et al., 2024; Bian et al., 2023), we introduce the context information to the attention mechanism. Specifically, we develop a Context-aware Cross-Attention (CCA) operation to optimize the key-aware deformable attention mechanism. Instead of using unstable point-level similarity, our method leverages patch-level similarity to compute the attention weights. The patch-level attention utilizes more context features to prevent the attention weights from being disturbed, thereby bridging the gap between object-level and point-level tasks. For *temporal feature querying*, to address the drifting issue, we discard the RNN-like long-temporal modeling in TAPTRv2. Instead, for any tracking point, we resort to the initial feature sampled from its starting frame, as this feature is the most reliable, and use it as input in any frame. To compensate for feature change over time, we introduce a Visibility-aware Long-Temporal Attention (VLTA) operation, which treats the initial feature as a query and performs dense attention over the past frames to aggregate past feature changes. This not only enables the perception of longer temporal context but also makes TAPTRv3 an online tracker. Meanwhile, recognizing that the target tracking point may be occluded in some frames, we reweight the long-temporal attention weights using the estimated visibility scores in the past frames, directing more attention to frames where the point is visible. This further enhances the feature querying ability along the temporal dimension.

For the scene cut issue, we introduce an auto-triggered global matching mechanism to reinitialize the point query's positional part for subsequent frames. Note that we only trigger the global matching when detecting a scene cut. This is based on our observation that in regular videos, using the predicted positions from the previous frame as the initial position for the current frame yields better results.

In summary, our contributions are threefold: (1) The primary contribution is the development of a more robust solution for point tracking in long videos, which improves upon TAPTRv2 by leveraging both spatial and temporal context. The two corresponding operations, namely Context-aware Cross-Attention and Visibility-aware Long-Temporal Attention, effectively improve the quality of spatial cross attention and long-term feature updating, enhancing feature querying. (2) To address the scene cut issue, we introduce an auto-triggered global matching mechanism, which is only triggered when a scene cut is detected. This ensures stable tracking on regular videos while being able to quickly

reestablish tracking when encountering scene cuts. (3) Extensive experimental results show that TAPTRv3 significantly outperforms TAPTRv2 and achieves state-of-the-art performance on most datasets. Even when compared to models trained on large-scale extra internal real data, TAPTRv3 remains competitive.

## 2 RELATED WORK

**Optical Flow Estimation**. Establishing the correspondences for every pixel between two consecutive frames is a long-standing problem. Over the past few decades, extensive research has been dedicated to addressing this issue. Traditional methods (Horn & Schunck, 1981; Black & Anandan, 2002; Bruhn et al., 2005) use carefully designed descriptors to find correspondences and apply manually designed rules to filter out distractions. DCFlow (Xu et al., 2017) first demonstrated the feasibility of using the features learned from deep neural network to obtain optical flow estimation through cost-volume that is constructed by calculating 4D correlation, and dominant this field (Teed & Deng, 2020; Dosovitskiy et al., 2015; Ilg et al., 2017; Xu et al., 2017; Sun et al., 2018; Wang et al., 2020; Jiang et al., 2021; Xu et al., 2021; Zhang et al., 2021; Huang et al., 2022; Zhao et al., 2022). Although the features extracted by deep neural networks are much stronger, the cost-volume still suffers from ambiguity. To address this, these methods typically feed the cost-volume into convolutions to normalize it based on contextual information. Although the recent optical flow estimation methods (Shi et al., 2023a; Saxena et al., 2024; Shi et al., 2023b) have shown remarkable results, they still can not handle video data well, especially when points of interest are occluded.

**Tracking Any Point**. Influenced by optical flow estimation methods, especially the RAFT (Teed & Deng, 2020), most approaches (Doersch et al., 2022; 2023; 2024; Zheng et al., 2023; Karaev et al., 2024b;a; Cho et al., 2024) follow a similar framework, calculating a cost-volume between the target tracking point and every frame, and then feeding the cost-volume into a transformer (Vaswani et al., 2017) to regress the position of the point in each frame. Inspired by the optical flow method (Teed & Deng, 2020), LocoTrack (Cho et al., 2024) introduces a local 4D correlation to enhance performance. TAG (Harley et al., 2024) extends tracking points to tracking arbitrary targets in videos. AnthroTAP (Kim et al., 2025) proposes a pipeline to generate training labels for point tracking from human motion data. Track-On (Aydemir et al., 2025) focuses on online tracking, introducing memory modules to capture temporal information for reliable point tracking. TAPNext (Zholus et al., 2025) is also an online model that casts this task as sequential masked token decoding and removes tracking-specific inductive biases. In another line, TAPTR and TAPTRv2 (Li et al., 2024c;b) address the TAP task from the perspective of detection with their DETR-like (Carion et al., 2020; Liu et al., 2022; Li et al., 2022; Zhang et al., 2023a) framework. However, TAPTRv2 still lacks designs for more challenging long-term tracking, resulting in suboptimal performance in long sequences.

## 3 METHOD

### 3.1 OVERVIEW

Before describing TAPTRv3, for clarity and without loss of generality, we assume that only a single point is being tracked, starting from the first frame $\mathbf{I}_0$. Given $\mathbf{I}_0$ and the user-specified point to be tracked at $\mathbf{l}_0 \in \mathbb{R}^2$, TAPTRv3 is expected to detect this point in every subsequent frame $\{\mathbf{I}_t\}_{t=1}^{T-1}$, determining its location $\{\mathbf{l}_t\}_{t=1}^{T-1}$ and visibility $\{\alpha_t\}_{t=1}^{T-1}$, where $\mathbf{l}_t \in \mathbb{R}^2$, $\alpha_t \in [0, 1]$, and $T$ is an integer that indicates the length of the video. As shown in Fig. 1 (a), TAPTRv3 can be roughly divided into the Point Query Preparation stage and the Sequential Point Tracking stage.

**Point Query and Spatial Context Preparation**. As depicted in Fig. 1 (a), given a user-specified coordinate $\mathbf{l}_0$ on the initial frame $\mathbf{I}_0$, the point query preparation stage will sample a point-level feature $\mathbf{f} \in \mathbb{R}^D$ to describe the target tracking point, where $D$ is the number of channels. Following TAPTRv2, we conduct bilinear interpolation on the $\mathbf{I}_0$'s corresponding image feature map[1] $\mathbf{X}_0 \in \mathbb{R}^{H \times W \times D}$ at $\mathbf{l}_0$, where $H$ and $W$ indicate the height and width of the feature map. To create a more comprehensive description of the point, TAPTRv3 additionally samples $N^2$ context features $\mathbf{C} \in \mathbb{R}^{N^2 \times D}$ around $\mathbf{l}_0$ in a grid form to describe the point's initial spatial context:

$$\mathbf{C} = \texttt{Bili}\left(\mathbf{X}_0, \mathbf{l}_0 + \mathbf{G}\right),\tag{1}$$

---

[1]For clarity, without loss of generality, we assume that each frame's feature map, obtained from the backbone and transformer encoder, has only one scale.

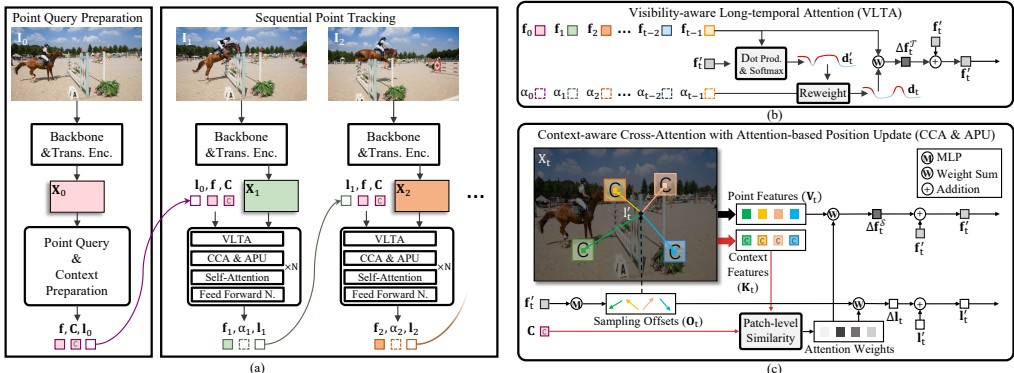

Figure 1: **Overview (a) and core components (b) (c) of TAPTRv3.** After the user specifies the point to track, the point query preparation stage prepares the content and spatial context features for this point in the initial frame. When TAPTRv3 receives a new frame, the sequential point tracking stage uses the content feature and the specified location as the point query, and regards the new frame's image feature map as keys and values. The points query, keys, and values are fed into a multi-layer transformer decoder to detect the tracking point in the new frame. The predicted location then updates the point query's positional part, providing a better initial position for tracking in the next frame. For clarity, the global matching module is not plotted.

where $N$ is a hyperparameter and is set as 3 by default, $\mathbf{G} \in \mathbb{R}^{N^2 \times 2}$ is the sampling grid, and `Bili` is the bilinear interpolation operation. After that, $\mathbf{f}$ and $\mathbf{l}_0$ will be regarded as the initial content part and the positional part of the target tracking point's corresponding point query. The point query will be sent to the transformer decoder as a query to detect the target tracking point in subsequent frames in the next Sequential Point Tracking stage.

**Sequential Point Tracking**. As shown in Fig. 1 (b), when TAPTRv3 receives a new frame, $\mathbf{I}_1$ for example, its corresponding feature map $\mathbf{X}_1$, which is regarded as a set of keys and values, as well as the point query, will be sent to the multi-layer transformer decoder. In each transformer decoder layer, both the content part and the positional part of the point query will be refined by our Visibility-aware Long-Temporal Attention, Context-aware Cross-Attention with APU, Self-Attention, and Feed Forward Network. After the multi-layer refinement. The output refined positional part of the point query $\mathbf{l}_1$ will be regarded as the detection result of the target tracking point in $\mathbf{I}_1$, and the output refined content feature of the point query $\mathbf{f}_1$ will be sent to an MLP-based binary classifier to predict the confidence $\alpha_1$ that the point is visible in $\mathbf{I}_1$. After that, the position prediction $\mathbf{l}_1$ will be used to update the point query's positional part, providing a better initial position for detecting the target tracking point in the next frame, while the content part remains as the initial content feature $\mathbf{f}$. This process proceeds repeatedly until the end of the video.

### 3.2 VISIBILITY-AWARE LONG-TEMPORAL ATTENTION

The RNN-like long-temporal modeling in TAPTR and TAPTRv2 can lead to feature drift, primarily due to the excessive feature updates during testing, rooted in the disparity in video lengths between training (24 frames) and testing (ranging from 50 to 1300 frames). Thus, as shown in Fig. 1 (b), we resort to the attention mechanism for its ability to handle varying token lengths, as demonstrated in modern LLMs (Zhang et al., 2025; 2023b; OpenAI, 2023; Google, 2023). Following modern LLMs, we also utilize the rotary positional embedding (Su et al., 2024) to help our long-temporal attention pay more attention to recent frames. More specifically, the long-temporal attention weights of the target tracking point between the $t$-th frame and all past frames[2] can be first computed as:

$$\mathbf{F}_t = [\mathbf{f}_0, \mathbf{f}_1, \cdots, \mathbf{f}_{t-1}]^\top, \mathbf{R}_t = [\mathbf{r}_0, \mathbf{r}_1, \ldots, \mathbf{r}_{t-1}]^\top,$$
$$\mathbf{d}'_t = \texttt{SoftMax}\left((\mathbf{F}_t + \mathbf{R}_t) \otimes (\mathbf{f}'_t + \mathbf{r}_t)\right), \tag{2}$$

where $\mathbf{F}_t \in \mathbb{R}^{t \times D}$ and $\mathbf{R}_t \in \mathbb{R}^{t \times D}$ are the point query's refined content features in the past frames of the $t$-th frame, and their corresponding rotary frame index embeddings, $\mathbf{d}'_t \in \mathbb{R}^t$ is the long-temporal

---

[2]In practice, it is not necessary to interact with all past frames, and this is only for ease of description. For more details, please refer to the Sec. A.2 in our appendix.

attention distribution (weights), $\otimes$ indicates the matrix multiplication, and $\mathbf{f}_t' \in \mathbb{R}^D$ is the content feature of the point query in the $t$-th frame that has not been fully refined by decoder. In the first decoder layer, $\mathbf{f}_t' \equiv \mathbf{f}$.

Different from the textual scenarios, since the target tracking point will be occluded sometimes, the refined content features from the frames in which the target tracking point is occluded may contribute noise to the long-temporal attention. To prevent being affected by the noise, we utilize the visibility predictions in the past frames $\mathbf{a}_t \in \mathbb{R}^t = [\alpha_0, \alpha_1, \ldots, \alpha_{t-1}]^T$ to reweight the attention distribution, making it visibility-aware. The final visibility-aware attention weights are used to weight-sum their corresponding content features to obtain the temporal querying result $\Delta \mathbf{f}_t^{\mathcal{T}} \in \mathbb{R}^D$. The querying result will be further utilized as a residual to update the point query's content feature to complete the VLTA. The process can be formulated as:

$$\mathbf{d}_t = \frac{\mathbf{d}_t' \odot \mathbf{a}_t}{\mathtt{Sum}(\mathbf{a}_t)}, \quad \Delta \mathbf{f}_t^{\mathcal{T}} = \mathbf{F}_t^\top \otimes \mathbf{d}_t, \quad \mathbf{f}_t' \Leftarrow \mathtt{LN}\left(\mathbf{f}_t' + \Delta \mathbf{f}_t^{\mathcal{T}}\right), \tag{3}$$

where $\mathbf{d}_t \in \mathbb{R}^t$ is the attention distribution that is reweighted by visibility predictions, $\mathbf{f}_t'$ is the output and will be sent to the following modules for further refinement, $\mathtt{LN}$ is the Layernorm (Lei Ba et al., 2016), and $\odot$ is the element-wise multiplication.

## 3.3 CONTEXT-AWARE CROSS-ATTENTION WITH APU

Unlike object-level DETR-like methods, the content feature of a point query in TAPTRv2 is a point-level feature. This granularity limits the model's receptive field during cross-attention with the image feature map, often causing ambiguity in attention weights. This issue becomes more serious when the target tracking point undergoes significant variations or when the image contains uniform regions or repetitive patterns. Such conditions can lead to noisy querying of spatial features as well as noisy position updates in the cross-attention's belonging APU block. Inspired by previous methods (Teed & Deng, 2020; Bian et al., 2023; Cho et al., 2024), we propose integrating richer spatial context into the attention mechanism. This provides point queries with a more comprehensive understanding of their surroundings, resulting in more accurate and robust attention weights.

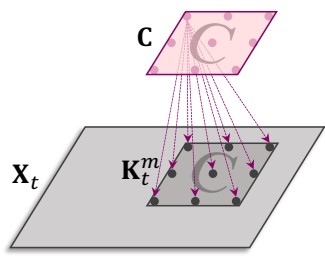

As illustrated in Fig. 1 (c), different from the vanilla key-aware deformable attention (Li et al., 2023), the query's point-level content feature will only be used to predict $M$ sampling offsets $\mathbf{O}_t \in \mathbb{R}^{M \times 2} = \left[\mathbf{o}_t^0, \mathbf{o}_t^1, \ldots, \mathbf{o}_t^{M-1}\right]^\top$ by an MLP. For the corresponding attention weights, we obtain them by leveraging the patch-level context features. More specifically, as shown in Fig. 2, for the $m$-th sampling point, its corresponding spatial context features in the $t$-th frame $\mathbf{K}_t^m \in \mathbb{R}^{N^2 \times D}$ can be constructed by:

Figure 2: **Illustration of patch-level similarity calculation.**

$$\mathbf{K}_t^m = \mathtt{Bili}\left(\mathbf{X}_t, \mathbf{l}_t' + \mathbf{o}_t^m + \mathbf{G}\right), \tag{4}$$

where $\mathbf{l}_t' \in \mathbb{R}^2$ is the positional part of the point query in the $t$-th frame that has not been fully refined by decoder layers. In the first decoder layer's CCA module $\mathbf{l}_t' \equiv \mathbf{l}_{t-1}$. After that, the $m$-th sampling point's corresponding patch-level similarity $w_t^m \in \mathbb{R}$ is calculated as:

$$\mathbf{S}_t^m \in \mathbb{R}^{N^2 \times N^2} = \mathbf{C} \otimes \mathbf{K}_t^{m\top}, \quad w_t^m = \mathtt{MLP}(\mathtt{Flatten}(\mathbf{S}_t^m)), \tag{5}$$

where $\mathbf{S}_t^m$ is the intermediate representation of the patch-level similarity, and $\mathtt{Flatten}$ indicates the flatten operation. By sending the intermediate representation $\mathbf{S}_t^m$ to an MLP, the more fine-grained similarities between every two points in the two patches are comprehensively considered, leading to a high-quality patch-level similarity. The similarities between the point query and all sampling points $\mathbf{w}_t \in \mathbb{R}^M = [w_t^0, w_t^1, \ldots, w_t^{M-1}]^\top$ will work as attention weights to aggregate their corresponding values $\mathbf{V}_t \in \mathbb{R}^{M \times D}$, obtaining the spatial querying result $\Delta \mathbf{f}_t^{\mathcal{S}} \in \mathbb{R}^D$:

$$\mathbf{V}_t = [\mathbf{v}_t^0, \mathbf{v}_t^1, \cdots, \mathbf{v}_t^{M-1}]^\top, \quad \mathbf{v}_t^m = \mathtt{Bili}(\mathbf{X}_t, \mathbf{l}_t' + \mathbf{o}_t^m),$$
$$\Delta \mathbf{f}_t^{\mathcal{S}} = \mathbf{V}_t^\top \otimes \mathtt{SoftMax}\left(\mathbf{w}_t/\sqrt{D}\right), \tag{6}$$

where $\mathbf{V_t}$ are the sampled values of $M$ sampling points on $\mathbf{X}_t$. After that, the spatial querying results will be utilized as a residual to complete the CCA:

$$\mathbf{f}_t' \Leftarrow \mathtt{LN}(\mathbf{f}_t' + \Delta \mathbf{f}_t^{\mathcal{S}}), \tag{7}$$

Following TAPTRv2, we also add the attention-based position update (APU) in CCA, to benefit from the attention weights' resilience to the domain gap without contaminating the point query. This process can be formulated as:

$$\Delta \mathbf{l}_t = \mathbf{O}_t^\top \otimes \texttt{SoftMax}\left(\texttt{MLP}\left(\mathbf{w}_t\right)/\sqrt{D}\right), \quad \mathbf{l}_t' \Leftarrow \mathbf{l}_t' + \Delta \mathbf{l}_t, \tag{8}$$

where $\Delta \mathbf{l}_t \in \mathbb{R}^2$ is the additional position update. We follow TAPTRv2, using an MLP to decouple the attention weights used for content and position updates.

## 3.4 Auto-Triggered Global Matching

In general, point motions are smooth in video. However, in long videos, especially in offline scenarios, scene cut often occurs, leading to sudden large motions. Since the initial position of a point query in the current frame is inherited from the last frame's prediction, as we have described in Sec.3.1, the sudden large motion will cost many frames for TAPTRv3 to catch up with the target tracking point.

Thus, when a scene cut occurs[3] at the $t$-th frame, we trigger the global matching module to help TAPTRv3 reestablish tracking. Similar to the previous method (Doersch et al., 2023), the global matching module will construct a similarity map $\mathbf{H}_t \in \mathbb{R}^{H \times W}$ between the target tracking point and the current frame's feature map. However, instead of relying on a point-level feature as in previous methods, similar to CCA, we leverage the spatial context features to help improve the accuracy of the similarity map. After that, SoftArgMax is conducted on the similarity map to obtain a final position prediction $\mathbf{l}_t \in \mathbb{R}^2$ that is differentiable. The prediction will be used to replace the unreliable position prediction of the current frame from the transformer decoder. The replacement helps TAPTRv3 to prevent wrong predictions in the subsequent frames. The process can be formulated as:

$$\mathbf{H}_t' = \mathbf{X}_t \otimes \mathbf{C}^\top, \mathbf{H}_t = \texttt{Softmax}(\texttt{MLP}(\mathbf{H}_t')),$$
$$\mathbf{l}_t = \texttt{SoftArgMax}(\mathbf{H}_t), \tag{9}$$

where $\mathbf{H}_t' \in \mathbb{R}^{H \times W \times N^2}$ is the similarity maps between the initial context features and the current frame's feature map. These maps will be fused through an MLP to obtain the $\mathbf{H}_t$.

To avoid misunderstanding, we emphasize that our main contribution in global matching lies in the novel auto-trigger mechanism, rather than the global matching itself. Compared to the predictions of previous frame from our model, the localization provided by global matching is less precise. However, its key advantage is the ability to quickly provide a coarse global position, and our approach successfully combines the strengths of both methods.

## 4 Experiments

### 4.1 Datasets and Evaluation

**Training Data**. For fair comparison, following the previous methods (Doersch et al., 2023; Karaev et al., 2024b; Li et al., 2024c;b), we trained our model on the TAP-Vid-Kubric (Doersch et al., 2022) dataset, which consists of 11,000 synthetic videos generated by Kubric Engine (Greff et al., 2022), each containing 24 frames showing 3D rigid objects falling to the ground and bouncing. Each video has 2,048 points sampled on moving objects and backgrounds to be tracked, and their corresponding trajectories are also generated for training. The points to be tracked in the video are occasionally occluded to allow the model to cope with this situation. During training, the resolution of the videos is resized to $384 \times 512$, and we randomly select 800 trajectories in each video for efficient training.

**Evaluation Data**. We follow previous methods to evaluate TAPTRv3 on the challenging TAP-Vid (Doersch et al., 2022) benchmark, which consists of 3 subsets, TAP-Vid-Kinetics, TAP-Vid-DAVIS, and RGB-Stacking. TAP-Vid-DAVIS comprises 30 real-world videos from the DAVIS 2017 validation set (Perazzi et al., 2016). These videos are relatively short, averaging less than 70 frames, so we include the experiments on this subset in Section A.3 of the appendix. TAP-Vid-Kinetics contains 1,144 YouTube videos from the Kinetics-700-2020 validations set (Carreira & Zisserman, 2017), with camera shakes, complex environments, and even scene cuts. These videos are relatively long, averaging about 250 frames per video. RGB-Stacking is a synthetic dataset that captures the process of a robotic arm grasping solid-colored blocks, with an average duration of about 250 frames. Although it has a relatively smaller domain gap compared to the training set, which is also a synthetic

---

[3]We use the off-the-shelf PySceneDetect (Castellano) to detect the scene cuts.

Table 1: **Comparison of TAPTRv3 with prior methods.** We use $^\dagger$ to indicate the introduction of additional training data. Specifically, CoTracker3$^\dagger$ additionally incorporates 15K real videos. BootsTAPIR$^\dagger$ and BootsTAPNext-B train on an additional 15M real videos. Anthro-LocoTrackR$^\dagger$ leverages an extra 1.4K human motion data. TAPTRv3 obtains state-of-the-art performance on most datasets and remains competitive with methods trained on extra internal data. Training data: (Kub24), (Kub48), and (Kub64) refer to Kubric (Greff et al., 2022) with 24, 48, and 64 frames per video, respectively. (PO) PointOdyssey (Zheng et al., 2023), (FT) FlyingThings++ (Mayer et al., 2016). For fair comparison, we do not utilize auto-triggered global matching here.

| Method | Training Data | Kinetics | | | RGB-Stacking | | | RoboTAP | | |
|---|---|---|---|---|---|---|---|---|---|---|
| | | AJ | $< \delta^x_{avg}$ | OA | AJ | $< \delta^x_{avg}$ | OA | AJ | $< \delta^x_{avg}$ | OA |
| PIPs (Harley et al., 2022) | FT | 31.7 | 53.7 | 72.9 | – | 59.1 | – | – | – | – |
| PIPs++ (Zheng et al., 2023) | PO | – | 63.5 | – | – | 58.5 | – | – | 63.0 | – |
| TAP-Net (Doersch et al., 2022) | Kub24 | 38.5 | 54.4 | 80.6 | 53.5 | 68.1 | 86.3 | 45.1 | 62.1 | 82.9 |
| TAPIR (Doersch et al., 2023) | Kub24 | 49.6 | 64.2 | 85.0 | 55.5 | 69.7 | 88.0 | 59.6 | 73.4 | 87.0 |
| CoTracker (Karaev et al., 2024b) | Kub24 | 49.6 | 64.3 | 83.3 | 67.4 | 78.9 | 85.2 | 58.6 | 70.6 | 87.0 |
| TAPTR (Li et al., 2024c) | Kub24 | 49.0 | 64.4 | 85.2 | 60.8 | 76.2 | 87.0 | 60.1 | 75.3 | 86.9 |
| TAPTRv2 (Li et al., 2024b) | Kub24 | 49.7 | 64.2 | 85.7 | 53.4 | 70.5 | 81.2 | 60.9 | 74.6 | 87.7 |
| LocoTrack (Cho et al., 2024) | Kub24 | 52.9 | 66.8 | 85.3 | 69.7 | 83.2 | 89.5 | 62.3 | 76.2 | 87.1 |
| CoTracker3 (online) (Karaev et al., 2024a) | Kub64 | 54.1 | 66.6 | 87.1 | 71.1 | 81.9 | 90.3 | 60.8 | 73.7 | 87.1 |
| Track-On (online) (Aydemir et al., 2025) | Kub24 | 53.9 | 67.3 | 87.8 | 71.4 | 85.2 | 91.7 | 63.5 | 76.4 | 89.4 |
| BootsTAPIR$^\dagger$ (Doersch et al., 2024) | Kub24+15M | 54.6 | 68.4 | 86.5 | 70.8 | 83.0 | 89.9 | 64.9 | 80.1 | 86.3 |
| CoTracker3 (online)$^\dagger$ (Karaev et al., 2024a) | Kub64+15K | 55.8 | 68.5 | 88.3 | 71.7 | 83.6 | 91.1 | 66.4 | 78.8 | 90.8 |
| Anthro-LocoTrack$^\dagger$ (Kim et al., 2025) | Kub24+1.4K | 53.9 | 68.4 | 86.4 | - | - | - | 64.7 | 79.2 | 88.4 |
| BootsTAPNext-B (online)$^\dagger$ (Zholus et al., 2025) | Kub48+15M | 57.3 | 70.6 | 87.4 | - | - | - | - | - | - |
| TAPTRv3 (online) | Kub24 | 54.9 | 67.5 | 88.2 | 72.3 | 84.1 | 90.8 | 64.5 | 77.3 | 89.7 |

dataset, the objects in this dataset often lack texture, making them difficult to track. In addition, we also evaluated TAPTRv3 on RoboTAP (Vecerik et al., 2024), which comprises 265 real-world videos from robotic manipulation tasks. The video length in this dataset varies significantly, with the longest videos reaching up to 1,300 frames and the average length of about 270 frames per video.

**Evaluation Metrics and Settings**. We use the standard metrics proposed in TAP-Vid (Doersch et al., 2022) for evaluation, including three important metrics. Occlusion Accuracy (OA), describes the accuracy of classifying whether a target tracking point is visible or occluded. $< \delta^x_{avg}$, reflecting the average precision of visible points' position at thresholds of 1, 2, 4, 8, and 16 pixels. Average Jaccard (AJ), a comprehensive metric, considers both the precision of position and visibility prediction. Since TAPTRv3 is an online tracker, we use the "First query" mode (Doersch et al., 2022) to evaluate the model, which tracks the target tracking point from the first frame when they are visible until the end of the video. This is much more difficult than the "Strided query" mode for offline trackers. Besides, since the resolution of the input video has a large impact on the final performance, we limit the resolution of the input video to $256 \times 256$ for a fair comparison with other methods during evaluation.

## 4.2 IMPLEMENTATION DETAIL

Unlike previous works (Li et al., 2024c;b), we use Resnet-18 instead of Resnet-50 as the backbone for higher efficiency. In the Transformer, we employ two encoder layers with deformable attention (Zhu et al., 2021) to further enhance the image features. While benefiting from our improvement, only 4 decoder layers are required to achieve optimal performance. For the supervision of location and visibility prediction, we utilize the L1 loss and binary cross-entropy loss, respectively, as in previous works. We use the AdamW (Loshchilov, 2017) optimizer with $\beta_1 = 0.9$ and $\beta_2 = 0.999$ and set the weight decay to $1 \times 10^{-4}$. We train TAPTRv3 on a cluster of 8 NVIDIA A100 GPUs for about 33,000 iterations with a batch size of 8 in total. To make the training process more stable, we accumulate gradients 4 times to approximate a batch size of 32. After the training of TAPTRv3, we freeze it and add the additional global matching for the second stage of training. Since there are only a few parameters to be trained in this stage, it only requires about 5,300 iterations to converge.

For efficiency, in ablation studies in Sec. 4.4, we have a few modifications in our experimental settings. We reduce the number of encoder and decoder layers to 1 and 3, respectively, resize the resolution of the input video to $384 \times 384$, and also reduce the number of tracking points on each video to 200. The ablation studies are conducted on 4 GeForce RTX3090 GPUs for about 33,000 iterations with randomly sampled half-size training and evaluation sets.

Table 2: **Ablations on key component of TAPTRv3.**
"LTA" refers to Long-Temporal Attention, "Vis-Aware" is short for visibility-aware, "Re. Win." indicates the removal of the sliding window, and "Sup. Vis." represents only supervising the visible points' positions during training.

| Row | LTA | Vis-Aware | Re. Win. | Sup. Vis. | CCA | AJ |
|-----|-----|-----------|----------|-----------|-----|------|
| 1 | ✗ | ✗ | ✗ | ✗ | ✗ | 44.5 |
| 2 | ✓ | ✗ | ✗ | ✗ | ✗ | 47.8 |
| 3 | ✓ | ✓ | ✗ | ✗ | ✗ | 48.8 |
| 4 | ✓ | ✓ | ✓ | ✗ | ✗ | 49.5 |
| 5 | ✓ | ✓ | ✓ | ✓ | ✗ | 51.1 |
| 6 | ✓ | ✓ | ✓ | ✓ | ✓ | **52.9** |

Table 3: **Ablation on patch-level similarity calculation.**

| Patch-level Similarity | AJ | $< \delta_{avg}^x$ | OA |
|------------------------|------|------|------|
| Element-wise | 51.3 | 64.8 | 87.4 |
| Every two point | **52.9** | **65.9** | **87.8** |

Table 4: **Ablation of methods for context features updating.**

| Update Methods | AJ | $< \delta_{avg}^x$ | OA |
|----------------|------|------|------|
| VLTA | 51.2 | 64.4 | 86.1 |
| MLP | 51.7 | 65.4 | 87.7 |
| No Updates | **52.9** | **65.9** | **87.8** |

## 4.3 Comparison with the State of the Arts

We evaluate TAPTRv3 on the Kinetics, RGB-Stacking, and RoboTAP datasets, which have relatively long videos, and compare it with previous methods to demonstrate its superiority on long videos. As shown in Table 1, TAPTRv3 achieves state-of-the-art on most metrics on these three datasets. With our insights and careful designs, TAPTRv3 shows a significant improvement (9.2 AJ on average) compared to TAPTRv2 (Li et al., 2024b) even with a more lightweight backbone and fewer decoder layers. Meanwhile, compared with the previous state-of-the-art Track-On (Aydemir et al., 2025) that uses DINOv2 (Oquab et al., 2023) as the backbone, we achieve an average improvement of 1.0 AJ.

Although Cotracker3 (Karaev et al., 2024a) and BootsTAPIR (Doersch et al., 2024) achieve remarkable performance, they both introduced extra internal real-world data for training. Specifically, CoTracker3 re-renders the Kubric training set with a length of 64 frames, which narrows the gap in video length between training and evaluation. Then, an additional 15K real-world videos are introduced for fine-tuning. BootsTAPIR trains its model on the original Kubric training set but introduces an extra 15M real-world video clips, which is approximately 1,360 times more than the synthetic data (11K) we use for training. The results show that despite these methods utilizing much more additional data for training, TAPTRv3 still achieves competitive performance. Note that, for fair comparison, we do not utilize auto-triggered global matching to help reestablish tracking here. When the global matching is also enabled, the results are further improved, as shown in Table 6 of Sec. 4.4.

## 4.4 Ablation Studies and Analysis

We start our ablation from TAPTRv2 and conduct ablation studies on every key component in TAPTRv3 to validate their effectiveness. We further conduct some more detailed ablations to investigate the best implementation choice. To focus on the ability of TAPTRv3 in handling long videos, we conduct ablations on TAP-Vid-Kinetics.

**Visibility-aware Long-Temporal Attention**. We first replace the RNN-like long-temporal modeling with the long-temporal attention. As shown in Table 2, the comparison between Row 2 and Row 1 shows that the replacement provides a large improvement (3.3 AJ), indicating the superiority of the attention mechanism over RNN in handling varying length, which aligns with the findings in modern LLMs. Furthermore, the comparison between Row 3 and Row 2 shows that enabling the long-temporal attention to utilize visibility prediction to reduce noises caused by occlusion will further improve the performance by 1.0 AJ. The significant improvements brought by the VLTA validate the effectiveness of incorporating richer temporal information.

**Removal of the Sliding Window**. With VLTA, the model captures temporal information from previous frames, making it redundant to recompute temporal attention within a small window. Therefore, we eliminate the sliding window by reducing the window size from 8 to 1. In this case, the original temporal attention in TAPTRv2 will degenerate to an MLP with residual connections. We retain this module in the following ablations for fair comparison. As shown in Table 2, the comparison between Row 4 and Row 3 shows a 0.7 AJ improvement. We attribute this to better position initialization. More specifically, initializing the position of the target tracking point in the current frame with the estimation from a nearby frame simplifies the estimation process. In addition, this modification also enables TAPTRv3 to process input videos in a streaming manner.

**Context-aware Cross-Attention**. As shown in Table 2, the comparison between Row 6 and Row 5 shows that the introduction of CCA further brings a significant improvement of 1.8 AJ, verifying the effectiveness of CCA in improving the robustness of the spatial feature querying.

Table 5: **Ablation on the input positions of decoder.**

| Input Positions of Decoder | AJ | $< \delta_{avg}^x$ | OA |
|---|---|---|---|
| Global Matching Calculation | 51.1 | 64.0 | 86.5 |
| Previous Frame's Prediction | **52.9** | **65.9** | **87.8** |

Table 6: **Ablation on global matching.** Whether to use auto-triggered global matching.

| Input Positions of Decoder | Dataset | AJ | $< \delta_{avg}^x$ | OA |
|---|---|---|---|---|
| Previous Frame's Prediction | Kin. | 54.9 | 67.5 | 88.2 |
| Global Matching if S.C. | Kin. | **55.1** | **67.7** | **88.4** |

**Only Supervise the Position of Visible Points**. It is worth noting that predicting the position of the target tracking point when it is occluded is an ill-posed problem. Forcing the model to localize the occluded point can destabilize the learning process and may lead the model to learn a bias toward a fixed motion pattern. As shown in Rows 5 and 4 of Table 2, simply ignoring the supervision of invisible points' location predictions results in an improvement of 1.6 AJ.

**Patch-level Similarity Calculation in CCA**. As shown in Table 3, we conduct comparative experiments on different methods to obtain patch-level similarity. The results show that the "Element-wise", which only considers the similarities between two points that are located at the same position in the two patches, is less effective than the "Every two point" strategy that is adopted in our current CCA. This is because the "Every two point" strategy has an advantage in handling more complex spatial variations, such as rotation, and is more tolerant of the sampling point's location. For more details, please refer to Sec. A.4 in our appendices.

**Spatial Context Updating**. As shown in Table 4, neither utilizing our VLTA nor an MLP to update the query's context features yields better results. This indicates that maintaining the target tracking points' spatial context throughout the tracking process not only reduces computation costs but also helps spatial feature querying. For more details, please refer to Sec. A.4 in our technical appendix.

**Ablation on Auto-Triggered Global Matching** As discussed in Sec. 3.4, TAPTRv3 triggers global matching only when a scene cut is detected. This design is based on the empirical finding that naively applying global matching at every frame results in inferior performance, as shown in Table 5. Therefore, TAPTRv3 defaults to using the previous frame's prediction for initialization. It activates global matching only when a scene cut is detected, which serves to re-establish tracking and prevent failure. This automatic trigger mechanism boosts the performance of our best model by 0.2 AJ, as shown in Table 6. We further construct a subset of Kinetics by selecting all videos with scene cuts. On this subset, this module brings an improvement of 0.5 AJ in the same experiment setting, confirming its effectiveness. More details can be found in Sec. A.1 of the appendix.

## 5 VISUALIZATION

We select a real-world video with 351 frames to showcase the improvement of our model. The video captures a very long train passing through the scene. At the beginning of the video, we generate query points using a $40 \times 40$ grid over the foreground region (excluding the sky). This example presents a typical long-term occlusion scenario of nearly 300 frames, posing a considerable challenge to model robustness. As shown in the Fig. 3, TAPTRv2 makes large-scale visibility mispredictions once the train enters the frame, and after the train leaves, its predicted point locations become unstable, with most tracks lost. In contrast, TAPTRv3 maintains stable and accurate tracking throughout the entire video, correctly predicts visibility during occlusion, and recovers accurate point locations once the train leaves the frame. This result highlights the effectiveness of our proposed components in mitigating feature drifting and achieving more accurate and robust tracking. Notably, for fair comparison, the global matching mechanism in TAPTRv3 was disabled.

## 6 CONCLUSION

In this paper, we have presented TAPTRv3, a strong method for the TAP task. TAPTRv3 improves TAPTRv2 primarily by developing the Context-aware Cross-Attention (CCA) and Visibility-aware Long-Temporal Attention (VLTA) to address the shortage of feature querying. CCA improves key-aware deformable attention by leveraging spatial context, which helps point-level tasks obtain more robust and accurate attention weights for updating both features and positions. VLTA replaces the RNN-like long-temporal modeling with an attention mechanism, enabling the perception of longer temporal context and mitigating the issue of feature drifting. VLTA further utilizes the visibilities to improve the quality of long-temporal attention, leading to a better feature querying ability along the temporal dimension. Additionally, TAPTRv3 further improves its performance in long videos

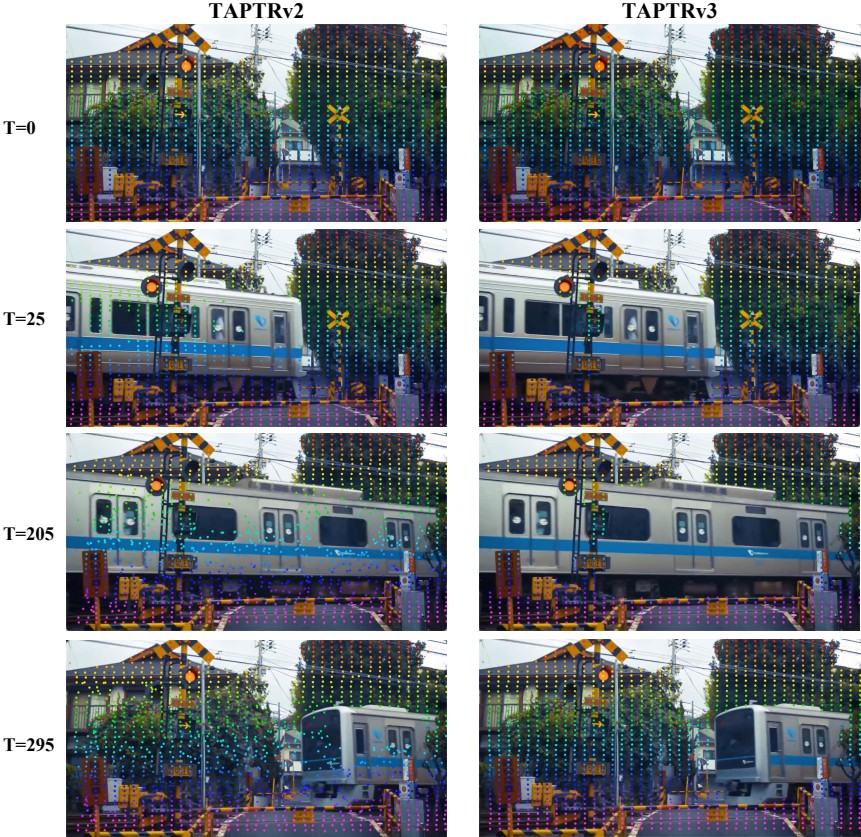

Figure 3: **Visual comparison between TAPTRv3 and TAPTRv2.**

by introducing the auto-triggered global matching mechanism. With the help of our insights and these novel designs, TAPTRv3 surpasses TAPTRv2 by a large margin and obtains state-of-the-art performance on multiple challenging datasets. Even when compared with the methods trained on extra internal data, TAPTRv3 remains competitive.

## ETHICS STATEMENT

Our work focuses on developing algorithms for tracking any points in videos, which is a fundamental computer vision problem with broad applications in robotics, video editing, and manipulation. This research relies exclusively on publicly available datasets and does not involve collecting or annotating data containing personal or sensitive information. All of our experiments are limited to standard academic benchmarks that are widely adopted in the community. We acknowledge that point tracking techniques could potentially be misused for surveillance or privacy-invasive applications. However, our intent is to advance the scientific understanding of visual correspondence and to enable positive downstream applications such as video editing, motion analysis, and embodied AI. We encourage responsible use of this technology in line with community norms and ethical guidelines.

## REPRODUCIBILITY STATEMENT

We have made every effort to ensure the reproducibility of TAPTRv3. Our method is described in detail in Sec.3, including the model design, architecture, and inference procedure. The training settings and hardware environment are provided in Sec.4.2. Additional implementation details and hyperparameter choices can be found in Sec.A.4 in the appendix. The datasets used for training and evaluation are introduced in Sec.4.1, all of them are publicly available and widely adopted benchmark datasets. All source code and model weights will be released to the public upon acceptance of this paper to further support research and development in the community.

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

# A    TECHNICAL APPENDICES AND SUPPLEMENTARY MATERIAL

In this appendix, we provide more ablation studies on the design of our key components and some hyper-parameters (see Sec. A.1). Beyond demonstrating the superior performance of our model, we also analyze its efficiency, including inference speed and GPU memory usage (see Sec. A.2). To comprehensively evaluate our model, we also conduct additional tests on the relatively short DAVIS dataset in TAP-Vid (Doersch et al., 2022) benchmark and the extremely long PointOdyssey Zheng et al. (2023) benchmark (see Sec.A.3). Furthermore, we provide more details on the implementation and experimental settings (see Sec.A.4). In addition, we also discuss the limitations of our model to provide a comprehensive understanding (see Sec.A.5). To address potential concerns, we also provide additional discussions on the model design (see Sec. A.6). In accordance with the requirements, we include a statement on the use of Large Language Models (see Sec. A.7). Finally, we present some more visualization results to show the superiority of TAPTRv3 and the effectiveness of our designs (see Sec. A.8 and the supplementary videos).

## A.1    MORE ABLATION STUDIES

**Number of Context Features**. As discussed in Sec. 3.1 and Sec. 3.3, we sample $N^2$ features in the form of a grid with 1-pixel grid interval when preparing the point query's context features $\mathbf{C}$ and each sampling point's context features $\mathbf{K}_t^m$ in cross-attention. At different scales of the feature map, the grid interval is the same. This allows us to fuse information from receptive fields of different sizes. As $N$ increases, it effectively means that the patch size becomes larger, and the context features tend to represent more global information. To find the optimal value of $N$, we conduct experiments with $N = 1$[4], $N = 3$, and $N = 5$ using the same settings as Sec. 4.4. The results are shown in Table 7, where $N = 3$ obtains the best performance. Compared with sampling 25 context features ($N = 5$), sampling 9 context features ($N = 3$) around the point not only requires fewer computing resources but also yields better performance, with an AJ improvement of 0.7. These results indicate that for the task of point tracking, context features are important, but they should not be so excessive or represent too large areas of the image as this could prevent accurate description of the tracked points.

Table 7: **Ablation on number of context features** $N^2$.

| Number of Context Features $N^2$ | AJ | $< \delta_{avg}^x$ | OA |
|:---:|:---:|:---:|:---:|
| $N^2 = 1$ | 51.3 | 64.6 | 87.4 |
| $N^2 = 9$ | **52.9** | **65.9** | **87.8** |
| $N^2 = 25$ | 52.2 | 65.8 | **87.8** |

**Memory Size of VLTA**. TAPTRv3 eliminates the sliding window (Li et al., 2024c;b) and introduces the Visibility-aware Long-Temporal Attention (VLTA) module to extend the temporal attention to an arbitrary length, while considering the visibility. Excluding the use of visibility, the recent SAM2 [5] adopts a similar temporal modeling approach, where SAM2 maintains a memory to record features from past frames in a FIFO manner and limits the memory size to 8 to focus on recent frames. To demonstrate the advantage of extending temporal attention to arbitrary lengths and the generalization ability of our VLTA to temporal lengths, we conduct ablation studies on the memory size. Specifically, we use the trained TAPTRv3 model from Sec 4.3 and perform evaluations on the Kinetics dataset with different memory sizes. As illustrated in Table 8, the significant positive correlation between memory size and model performance indicates that, although the perception range of our VLTA is limited to 1 to 23 frames during training [6], our VLTA is still able to generalize beyond 24 frames. This generalization ability enables us to expand the range of temporal perception, allowing the collection of long-term temporal information from all past frames to help improve the robustness of long-term point tracking.

---

[4]When $N = 1$, the CCA will actually degenerate to the vanilla key-aware deformable attention, we can also find this performance in Row 5 of Table 2.

[5]Nikhila Ravi, Valentin Gabeur, Yuan-Ting Hu, Ronghang Hu, Chaitanya Ryali, Tengyu Ma, Haitham Khedr, Roman R¨adle, Chloe Rolland, Laura Gustafson, et al. SAM 2: Segment Any-thing in Images and Videos. arXiv preprint arXiv:2408.00714, 2024. 12.

[6]Because our training data are videos with fixed lengths of 24 frames

Table 8: **Ablation on the memory size of VLTA (Kinetics).**

| Memory Size of VLTA | AJ | $< \delta_{avg}^x$ | OA |
|:---:|:---:|:---:|:---:|
| 12 | 51.9 | 65.2 | 86.5 |
| 24 | 53.1 | 66.3 | 87.3 |
| 48 | 54.5 | 67.2 | 88.0 |
| All Past | **54.9** | **67.5** | **88.2** |

However, unconstrained temporal memory size will lead to CUDA out-of-memory problems in practical applications. We find that when processing videos with up to 3,500 frames, the GPU memory requirement reaches 24GB because of the large amount of cached temporal memory. To balance performance and efficiency, and to enable the model to handle videos of arbitrary length, we need to limit the size of the temporal memory and manage it with FIFO mechanism. To this end, we further evaluated RoboTAP (with videos up to 1,300 frames) with more diverse memory sizes. As shown in Fig. 4, the performance converges when the memory size reaches 512. With the help of temporal memory management, TAPTRv3 is capable of handling downstream applications, which usually require the model to process online streaming videos.

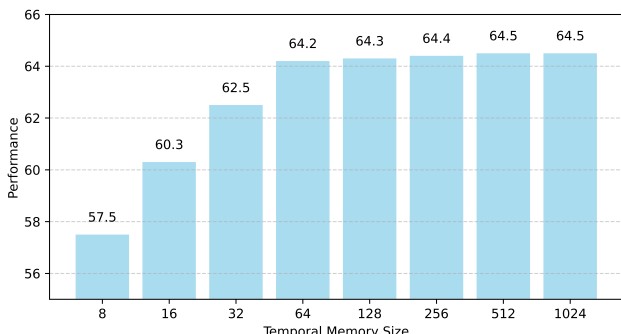

Figure 4: **Ablation on the memory size of VLTA (RoboTAP).**

**The Method of Obtaining Similarity Map in Global Matching**. In Sec. 3.4, we propose the auto-triggered global matching to reestablish point tracking when a scene cut is encountered to prevent the loss of tracking targets. Instead of simply relying on a point-level feature to construct a similarity map in global matching as in previous works (Cho et al., 2024; Doersch et al., 2023), we propose incorporating spatial context features, similar to the CCA module, to enhance accuracy. (For more implementation details, please refer to Sec. A.4). To investigate the effectiveness of this approach, we conduct an ablation study on it. As shown in Table 9, the incorporation of spatial context features brings a relatively small performance improvement. However, considering that we only trigger global matching when a scene cut is detected and its computational cost is negligible, we adopt the use of spatial context features in TAPTRv3.

Table 9: **Ablation on calculation of similarity map.**

| Feature of Tracking Point | AJ | $< \delta_{avg}^x$ | OA |
|:---:|:---:|:---:|:---:|
| Point-level Feature | 55.75 | 67.88 | **87.57** |
| Spatial Context Features | **55.79** | **67.94** | **87.57** |

To avoid misunderstanding, we emphasize that our main contribution in global matching lies in the novel auto-trigger mechanism, rather than the global matching itself.

**More Ablation on Auto-Triggered Global Matching**.

We evaluate the proposed auto-triggered global matching mechanism on the full Kinetics dataset. As shown in Table 6, while the method yields performance gains, the improvement is relatively modest. This is largely because only a subset of videos in Kinetics contain scene cuts, and thus actually activate the mechanism, whereas evaluation metrics are averaged over the entire dataset. To better isolate the effectiveness of our approach, we construct a scene-cut subset comprising all videos in Kinetics that contain at least one scene cut, accounting for approximately 27% of the 1,144 total videos. We conduct the same ablation study on this subset. As shown in Table 10, triggering global matching upon detecting scene cuts leads to a significant improvement (0.5 AJ), demonstrating the effectiveness of our method in relevant scenarios.

Table 10: **Ablation on auto-triggered global matching.** Whether to trigger the global matching when encountering scene cuts on the Kinetics scene-cut subset. "Kin." is short for TAP-Vid-Kinetics, and "S.C." indicates scene cuts

| Input Positions of Decoder | Dataset | AJ | $< \delta^x_{avg}$ | OA |
|---|---|---|---|---|
| Previous Frame's Prediction | Kin. w/ S.C. | 55.3 | 67.0 | 86.9 |
| Global Matching if S.C. | Kin. w/ S.C. | **55.8** | **67.9** | **87.6** |

## A.2 EFFICIENCY ANALYSIS

**Inference Speed**. Notably, while TAPTRv3 introduces two additional modules, CCA and VLTA, which increase computational overhead compared to TAPTRv2, the inference speed of TAPTRv3 is even faster than that of TAPTRv2. We benchmark the runtime performance on the TAP-Vid-DAVIS dataset using a single GeForce RTX3090 GPU, measuring the total number of video frames and the model's inference time to calculate the average FPS. We also include CoTracker (Karaev et al., 2024b), which is a highly representative and influential method, as a baseline. For all experiments, we set the input video resolution to $384 \times 512$, with an average of 22 tracked points per video. Each experiment was repeated five times to report a stable average. For a fair comparison, we set the window size to 8 and the window stride to 4 for all models. As shown in Table 11, under the same conditions, TAPTRv3 achieves 15 FPS higher than TAPTRv2.

Table 11: **Average FPS of TAPTRv2 and TAPTRv3**.

| Method | Average FPS | Method | Average FPS | Method | Average FPS |
|---|---|---|---|---|---|
| Cotracker | 26.4 | TAPTRv2 | 41.9 | TAPTRv3 | **57.2** |

We attribute the faster speed to the following reasons. (1) The CCA module introduces only 9 more sparse sample points, with the additional computational overhead negligible compared to the overall network. (2) While we introduce VLTA, we also remove the Temporal Attention module and the complex Window Post-processing in TAPTRv2. (3) TAPTRv3 uses a more lightweight backbone (Resnet-18 vs. Resnet-50) and fewer decoder layers (4 layers vs. 5 layers), making the model more efficient. Additionally, it utilizes lower-resolution training data ($384 \times 512$ vs. $512 \times 512$), resulting in lower-resolution inputs during inference as well.

**GPU Memory Overhead**. When the memory size in VLTA is set to 512 (as discussed in Sec A.1), our model is capable of processing videos of arbitrary length in a streaming manner. When tracking 100 points simultaneously in a streaming video, the GPU memory usage is less than 2GB, making its deployment cost-effective.

## A.3 MORE COMPARISONS

**Comparisons on DAVIS Benchmark.** We compare TAPTRv3 with previous methods on the DAVIS subset of the TAP-Vid (Doersch et al., 2022) benchmark. DAVIS contains relatively short videos, with an average length of only 70 frames. As shown in Table 13, on shorter videos TAPTRv3 performs comparably to TAPTRv2. Although it is slightly behind in AJ, there is still an improvement in $< \delta^x_{avg}$, indicating better localization capability. The result is reasonable, as our design is mainly intended to

Table 12: **Comparison of TAPTRv3 with different backbone and training resolution.**

| Method | Kinetics | | | RGB-Stacking | | | RoboTAP | | | DAVIS | | |
|---|---|---|---|---|---|---|---|---|---|---|---|---|
| | AJ | $< \delta_{avg}^x$ | OA | AJ | $< \delta_{avg}^x$ | OA | AJ | $< \delta_{avg}^x$ | OA | AJ | $< \delta_{avg}^x$ | OA |
| TAPTRv3 (Resnet-50, $512 \times 512$) | 54.5 | **67.5** | **88.2** | **73.0** | **86.2** | 90.0 | **64.6** | 77.2 | **90.1** | **63.2** | **76.7** | **91.0** |
| TAPTRv3 (Resnet-18, $384 \times 512$) | **54.9** | **67.5** | **88.2** | 72.3 | 84.1 | **90.8** | 64.5 | **77.3** | 89.7 | **63.2** | 76.4 | 90.6 |

Table 13: **Comparison of TAPTRv3 with prior methods.** We use $\dagger$ to indicate the introduction of additional training data. Specifically, CoTracker3$^\dagger$ additionally incorporates 15K real videos. BootsTAPIR$^\dagger$ and BootsTAPNext-B train on an additional 15M real videos. Anthro-LocoTrackR$^\dagger$ leverages an extra 1.4K human motion data. TAPTRv3 obtains state-of-the-art performance on most datasets and remains competitive with methods trained on extra internal data. Training data: (Kub24), (Kub48), and (Kub64) refer to Kubric (Greff et al., 2022) with 24, 48, and 64 frames per video, respectively. (PO) PointOdyssey (Zheng et al., 2023), (FT) FlyingThings++ (Mayer et al., 2016). For fair comparison, we do not utilize auto-triggered global matching here.

| Method | Training Data | DAVIS | | |
|---|---|---|---|---|
| | | AJ | $< \delta_{avg}^x$ | OA |
| PIPs (Harley et al., 2022) | FT | 42.2 | 64.8 | 77.7 |
| PIPs++ (Zheng et al., 2023) | PO | – | 69.1 | – |
| TAP-Net (Doersch et al., 2022) | Kub24 | 33.0 | 48.6 | 78.8 |
| TAPIR (Doersch et al., 2023) | Kub24 | 56.2 | 70.0 | 86.5 |
| CoTracker (Karaev et al., 2024b) | Kub24 | 61.8 | 76.1 | 88.3 |
| TAPTR (Li et al., 2024c) | Kub24 | 63.0 | 76.1 | 91.1 |
| TAPTRv2 (Li et al., 2024b) | Kub24 | 63.5 | 75.9 | **91.4** |
| LocoTrack (Cho et al., 2024) | Kub24 | 63.0 | 75.3 | 87.2 |
| CoTracker3 (online) (Karaev et al., 2024a) | Kub64 | 64.5 | 76.7 | 89.7 |
| Track-On (online) (Aydemir et al., 2025) | Kub24 | **65.0** | **78.0** | 90.8 |
| BootsTAPIR$^\dagger$ (Doersch et al., 2024) | Kub24+15M | 61.4 | 73.6 | 88.7 |
| CoTracker3 (online)$^\dagger$ (Karaev et al., 2024a) | Kub64+15K | 63.8 | 76.3 | 90.2 |
| Anthro-LocoTrack$^\dagger$ (Kim et al., 2025) | Kub24+1.4K | 64.8 | 77.3 | 89.1 |
| BootsTAPNext-B (online)$^\dagger$ (Zholus et al., 2025) | Kub48+15M | 65.2 | 78.5 | 91.2 |
| TAPTRv3 (online) | Kub24 | 63.2 | 76.4 | 90.6 |

improve tracking performance on long videos, which has already been validated in Table 1 of the main paper.

**Comparisons on PointOdyssey Benchmark.** PointOdyssey Zheng et al. (2023) is a benchmark composed of synthetic data, and it is noticeably more realistic than TAP-Vid-Kubric. A key characteristic of PointOdyssey is its extremely long video sequences: the average length in the test set reaches 2386 frames, and the longest video contains 4325 frames, which far exceeds the video lengths in the TAP-Vid benchmark. Since our method is designed to address challenges in long videos, evaluation on PointOdyssey is highly meaningful.

To this end, we directly evaluated TAPTRv3 trained only on the Kubric dataset on the PointOdyssey test set (including 12 valid videos). This allows us to assess the model's performance on longer and more challenging sequences, while also providing insight into its generalization ability. We report the metrics proposed in PointOdyssey, including $\delta_{avg}$, $\delta_{avg}^{vis}$, and $\delta_{avg}^{occ}$, which measure localization accuracy. The metrics $\delta_{avg}^{vis}$ and $\delta_{avg}^{occ}$ are computed in the same way as $\delta_{avg}$ but consider only visible and occluded points, respectively. In addition, we also report the Survival metric, which measures the average number of frames before tracking failure, where failure is defined as an error exceeding 50 pixels. The results are shown in the Table 14.

Even without being trained on PointOdyssey, TAPTRv3 achieves highly competitive performance. More importantly, the improvement compared to TAPTRv2 is extremely obvious, especially a 9.0 point gain on the $\delta_{avg}^{vis}$ metric, which measures tracking accuracy on visible points. The experimental result validates that our insight into the limitations of TAPTRv2 in long videos is correct, and our proposed modules are highly fruitful.

Table 14: **Comparison of TAPTRv3 with prior methods on PointOdyssey.**

| Method | Training Data | PointOdyssey $\delta_{avg}$ | $\delta_{avg}^{vis}$ | $\delta_{avg}^{occ}$ | Survival |
|---|---|---|---|---|---|
| TAP-Net (Doersch et al., 2022) | PointOdyssey | 28.4 | - | - | 18.3 |
| PIPs (Harley et al., 2022) | PointOdyssey | 27.3 | - | - | 42.3 |
| PIPs++ (Zheng et al., 2023) | PointOdyssey | 29.0 | 32.4 | 18.8 | 47.0 |
| CoTracker (Karaev et al., 2024b) | PointOdyssey | 30.2 | 32.7 | 24.2 | **55.2** |
| Track-On (Aydemir et al., 2025) | Kubric | **34.2** | **38.1** | - | 49.5 |
| TAPTRv2 (Li et al., 2024b) | Kubric | 26.1 | 28.4 | 21.5 | 50.0 |
| TAPTRv3 | Kubric | 33.9 | 37.4 | **26.0** | 51.3 |

## A.4 MORE IMPLEMENTATION DETAILS

**Backbone and Training Resolution**.

As discussed in Sec. 4.1 and Sec. 4.2, compared to TAPTR and TAPTRv2, which use Resnet-50 as the backbone to extract image features, TAPTRv3 adopts the more lightweight Resnet-18. Additionally, TAPTRv3 is trained on videos with a resolution of $384 \times 512$, which is lower than the $512 \times 512$ resolution used by TAPTR and TAPTRv2. This is primarily because our experiments showed that a lighter backbone and lower training resolution can achieve comparable performance, as indicated by the results in Table 12. These modifications improve the efficiency of our model and also demonstrate the superiority of the decoder design.

**Calculation of Similarity Map in Global Matching**. In the global matching module of TAPTRv3, we construct a similarity map $\mathbf{H}_t \in \mathbb{R}^{H \times W}$ between the target tracking point and the feature map $\mathbf{X}_t \in \mathbb{R}^{H \times W \times D}$ of the current frame. However, instead of solely relying on a point-level feature as in previous methods, similar to CCA, we leverage the spatial context features $\mathbf{C} \in \mathbb{R}^{N^2 \times D}$ to help improve the accuracy of the similarity map $\mathbf{H}_t$. As illustrated in Fig. 5, in our global matching, we first utilize each feature of the target tracking point's context feature to compute a group of similarity maps $H'_t \in \mathbb{R}^{H \times W \times N^2}$. However, since each feature in the context features is still point-level, computing the similarity map using any single one of them independently still introduces noise. Therefore, we further employ an MLP to comprehensively integrate these noisy similarity maps, resulting in a more accurate one $\mathbf{H}_t$.

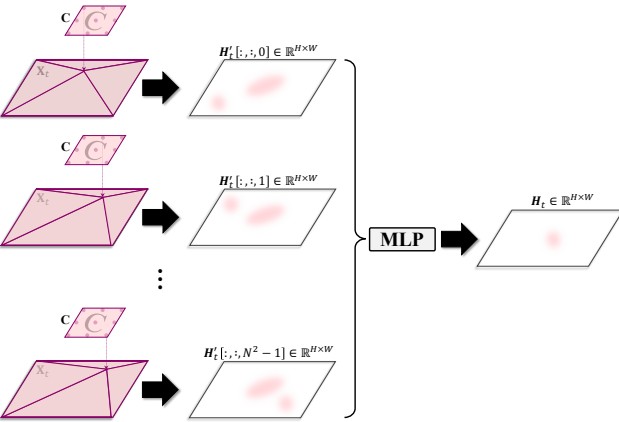

Figure 5: **Detailed visual illustration of global matching.**

**Patch-level Similarity Calculation in CCA**. Sec. 3.3 briefly introduces how to calculate patch-level similarity in Context-aware Cross-Attention (CCA), and Sec. 4.4 presents an ablation study on it. In this section, we provide a more detailed description. For the "Every two points" method that is adopted in TAPTRv3 by default, as shown in Fig 6 (a), each feature of the point query's $N^2$ context features is paired with every feature in the sampling point's $N^2$ context features $K_t^m$ to obtain

$\mathbf{S}_t^m \in \mathbb{R}^{N^2 \times N^2}$. For the "Element-wise" method (See Sec. 4.4 and Table 9), as shown in Fig 6 (b), each feature of the point query's $N^2$ context features is only paired with its corresponding one in $K_t^m$ to obtain $\mathbf{S}_t^m \in \mathbb{R}^{N^2}$

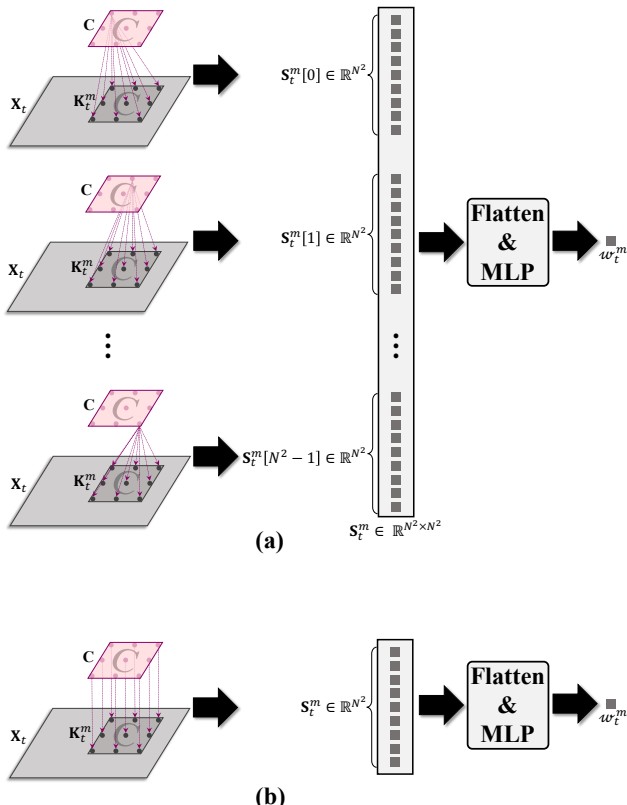

Figure 6: **Detailed visual illustration of different methods for computing patch-level similarity.**

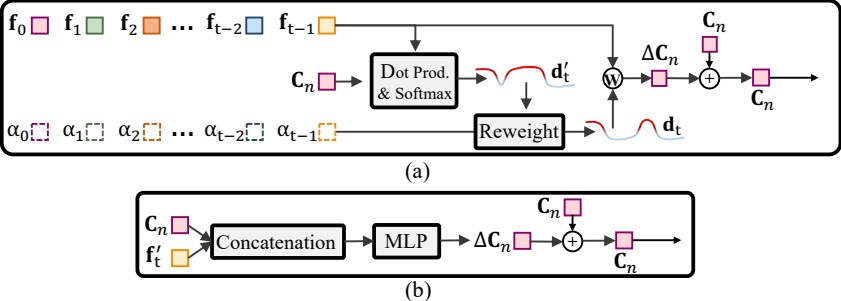

Figure 7: **Detailed visual illustration of different methods for the updating of spatial context features.**

**Spatial Context Updating**. In TAPTRv3, we consistently employ the initial spatial context features $\mathbf{C}$ throughout all decoder layers. To validate its effectiveness, we add a spatial context feature update module in the transformer decoder for comparison in our ablation study section (See Sec. 4.4 and Table 4). Here, we provide a detailed description of the two methods for updating spatial context features proposed in this ablation. For the one that uses our VLTA, as shown in Fig. 7 (a), for the $n$-th spatial context feature $\mathbf{C}_n$, we treat it as a query and attend VLTA to the refined content features of the past frames to update it. For the one that uses an MLP, as shown in Fig. 7 (b), we concatenate

the spatial context feature with the point query's content feature in the current frame $\mathbf{f}'_t$ and update the spatial context features with the MLP.

**Backbone and Number of Encoder Layers**. We further conduct experiments on the TAP-Vid Doersch et al. (2022) benchmark to investigate the influence of the backbone and encoder layers on model performance, reporting the Average Jaccard (AJ) metric. The results are presented in Table 15.

First, we employ a larger-capacity backbone, ViT-Adapter Chen et al. (2022) with DINOv2, and report the results in Row 3 of Table 15. Compared with ResNet18 (Row 2) and ResNet50 (Row 1), the performance decreases on all datasets, with a particularly notable drop on the synthetic RGB-Stacking dataset. Unlike methods based on other architectures, we do not observe that using a more advanced backbone in our DETR-like model yields further performance gains.

Table 15: **The impact of different backbone and encoder layers on model performance**

| Row | Method | Backbone | Enc. Layers | Kinetics AJ | RGB-Stacking AJ | DAVIS AJ |
|---|---|---|---|---|---|---|
| 1 | TAPTRv3 (Ours) | ResNet18 | 2 | **54.9** | 72.3 | 63.2 |
| 2 | TAPTRv3 (Ours) | ResNet50 | 2 | 54.5 | **73.0** | 63.2 |
| 3 | TAPTRv3 (Ours) | ViT-Adapter with DINOv2 | 2 | 54.2 | 67.8 | 62.2 |
| 4 | TAPTRv3 (Ours) | ResNet18 | 1 | 53.5 | 66.7 | 61.6 |
| 5 | TAPTRv3 (Ours) | ResNet18 | 3 | 51.9 | 62.6 | 58.7 |
| 6 | TAPTRv2 | ResNet18 | 2 | 45.6 | 46.7 | 60.0 |
| 7 | TAPTRv2 | ResNet50 | 2 | 49.7 | 53.4 | **63.5** |

Furthermore, since our model utilizes a DETR-like structure, it includes encoder layers in addition to the backbone to further aggregate features. To better understand this behavior in our model, we additionally conduct experiments examining the impact of encoder layers and summarize the results in Rows 1, 4, and 5 of Table 15. When using two encoder layers, our model achieves the best performance, and the impact of encoder depth is substantially larger than that of the backbone. We believe that using only a single encoder layer may lead to insufficient feature extraction, making the model capacity too limited relative to the task difficulty. In contrast, using three encoder layers noticeably slows convergence and increases training difficulty. It is worth noting that replacing the backbone or adjusting encoder depth typically requires extensive hyperparameter tuning (e.g., learning rate and weight decay) to achieve optimal performance. We did not perform extensive hyperparameter tuning here and leave it for future exploration.

Finally, we reduce the model size of TAPTRv2 by replacing the ResNet50 backbone with ResNet18, following the design choice in our TAPTRv3. As shown in Rows 6 and 7 in Table 15, the model exhibits a significant performance drop with the lighter backbone, which, to some extent, reflects the effectiveness of our newly introduced components in the decoder.

## A.5 LIMITATIONS

TAPTRv3 is an online tracker that can process streaming video input, meaning that the decoder handles only one frame at a time. This design leads to very low computational cost (see Sec. A.2) but comes with a limitation in parallelism. Even with sufficient GPU memory, the model can not fully utilize it. As a result, TAPTRv3 may appear slower than offline methods when evaluated in offline settings, since those methods can process multiple frames simultaneously in a single forward pass, while TAPTRv3 requires multiple passes. However, TAPTRv3 can still be adapted for offline use by simply applying a sliding window strategy with an increased window size to improve parallelism. While this adjustment may incur a slight performance drop, it provides a flexible trade-off between speed and accuracy depending on the application scenario.

## A.6 MORE DISCUSSIONS

**Discussion of Using Attention in VLTA**. Recently, readily applicable RNN-like architectures such as Mamba Gu & Dao (2024) and RWKV Peng et al. (2023) have demonstrated strong potential and efficiency. However, we choose to use attention rather than these more advanced RNN-like structures in the VLTA module based on the following considerations.

One of our core insights is that the information from the initial frame should serve as the most reliable anchor and be preserved throughout tracking. Although Mamba and RWKV improve long-range information propagation through state compression and selective forgetting, they fundamentally rely on a single evolving hidden state to carry information over time. Regardless of how efficient the mechanism is, this hidden state represents a lossy compression of the entire history. In long videos, a point may disappear for hundreds of frames, during which the fine-grained appearance details stored in the hidden state may degrade or become contaminated. In contrast, our approach keeps the initial-frame anchor intact and uses attention to directly retrieve information from historical frames to accommodate appearance changes. Visibility prediction is further incorporated to ensure that more reliable historical information is aggregated.

In summary, we believe that for the video point tracking task, hidden-state evolution is less suitable for handling the complex challenges of long videos. While attention may incur higher computational cost, by appropriately controlling the size of the historical buffer, our model achieves a good balance between performance and efficiency. As described in Sec. A.2, TAPTRv3 can process arbitrarily long videos in a streaming manner (tracking 100 points simultaneously with less than 2GB GPU memory), making it deployment cost-efficient.

**Discussion of the Effectiveness of Visibility-aware Attention**. As described in the main text, we apply the predicted visibility in a soft weighting manner, where frames predicted as more likely to be invisible are down-weighted according to confidence rather than discarded. When visibility predictions are accurate, this weighting is naturally beneficial. When predictions are incorrect, two cases may occur. If a visible point is predicted as invisible, the attention weights for these frames are reduced, leading the model to extract less historical information from them. As a result, the model relies more on the earlier frames where the point is visible, and these frames provide reliable information. If an invisible point is predicted as visible, the behavior degenerates to not using visibility weighting, which may introduce unreliable historical information. In summary, the visibility-aware attention mechanism provides an overall positive effect, as confirmed by the results in Table 2.

**Discussion of Scene Cut Detector**. Currently, our model relies on an external library as an independent module to detect scene cuts. We find that this traditional detection method is sufficiently accurate in most cases. As reported in Table 10, on a Kinetics subset consisting of roughly 300 complex real-world videos, the auto-trigger global matching mechanism yields a clear performance gain. More importantly, this setup allows us to validate our key intuition: compared with relying on the model's prediction from the previous frame, the position estimated by global matching is less accurate, but its advantage lies in providing a fast, coarse global localization. The two components should therefore be combined.

Nevertheless, the current approach remains imperfect. Using the scene cut detector as an independent, untrained module may lead to potential generalization issues. However, this is primarily a temporary compromise because we are currently constrained by available training datasets, making it difficult to integrate it into the overall framework for joint training. The Kubric dataset commonly used for this task contains synthetic data with a relatively singular domain. Training a model solely on this dataset to explicitly or implicitly determine whether a frame contains a scene cut is highly challenging. We believe that, with access to larger and more diverse training datasets in the future, it will become feasible to integrate scene cut detection into the overall framework for joint training, leading to improved performance and generalization.

## A.7 THE USE OF LARGE LANGUAGE MODELS

In this paper, we only use Large Language Models (LLMs) for translation and text polishing. No aspects related to model design, experimental design, or result analysis involve the use of LLMs. We believe that the use of LLMs in this paper does not affect its scientific contributions.

## A.8 MORE VISUALIZATIONS

To further demonstrate the superiority of TAPTRv3 and the effectiveness of our designs, we provide more visualizations of TAPTRv3's predictions on some challenging real-world long videos. In these visualizations, we label the frame ID at the upper left corner of each frame to indicate the timestamp (except Fig. 8). To make the tracking results more visually distinct, we adjust the color of the tracking points based on the overall color of the video.

**More Visualizations for Motion Trajectories**. We provide more visualizations on videos from DAVIS dataset, which is a well-received benchmark. As illustrated in Fig. 8, we visualize the complete trajectories produced by TAPTRv3 to demonstrate the consistency of the estimated motion.

**More Visual Comparison with TAPTRv2**. We provide more visualizations of the comparisons between TAPTRv2 and TAPTRv3, demonstrating the superiority of TAPTRv3. For more details, please refer to the image captions and the corresponding videos. The correspondences between images and videos are:

Fig. 9 ⇒ CompareVideo1_TAPTRv2.mp4 & CompareVideo1_TAPTRv3.mp4.

Fig. 10 ⇒ CompareVideo2_TAPTRv2.mp4 & CompareVideo2_TAPTRv3.mp4.

Fig. 11 ⇒ CompareVideo3_TAPTRv2.mp4 & CompareVideo3_TAPTRv3.mp4.

**Visual Comparison w. and wo. Global Matching**. We provide more visualizations of the comparisons between TAPTRv3 with and without the auto-triggered global matching, demonstrating its effectiveness. For more details, please refer to the image captions and the corresponding videos. The correspondences between images and videos are:

Fig. 12 ⇒ CompareVideo4_TAPTRv3wGM.mp4 & CompareVideo4_TAPTRv3woGM.mp4.

Fig. 13 ⇒ CompareVideo5_TAPTRv3wGM.mp4 & CompareVideo5_TAPTRv3woGM.mp4.

**More Robust Prediction Visualizations**. We additionally provide more visualizations to demonstrate TAPTRv3's robustness in in-the-wild scenarios, showing its potential for various downstream applications. The correspondences between images and videos are:

Fig. 14 (a) ⇒ RobustVideo1.mp4

Fig. 14 (b) ⇒ RobustVideo2.mp4.

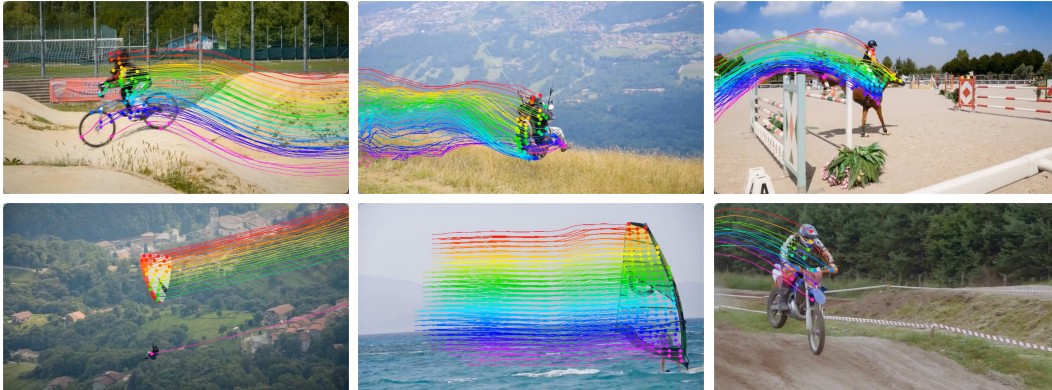

Figure 8: **More Visualizations for Motion Trajectories.**

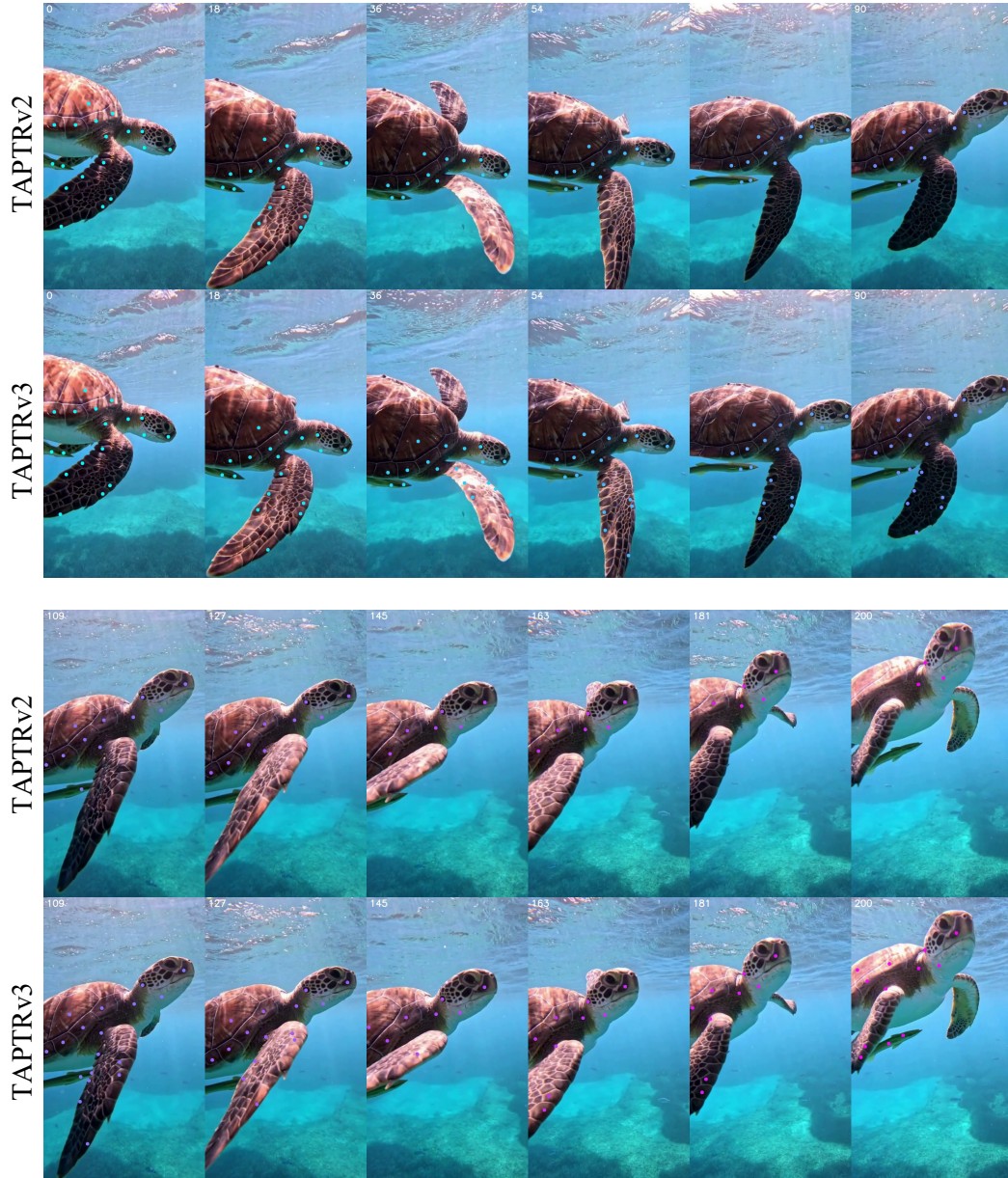

Figure 9: **Visual comparison between TAPTRv2 and TAPTRv3.** Best view in electronic version. From the third image in the first row (36th frame), TAPTRv2 loses tracking of the turtle's flippers, and in the last few frames loses tracking of the turtle shell and the point on the fish below the turtle. TAPTRv3, on the other hand, maintains stable and accurate tracking throughout the video. The corresponding videos (CompareVideo1_TAPTRv2.mp4 and CompareVideo1_TAPTRv3.mp4) are provided in the supplementary material.

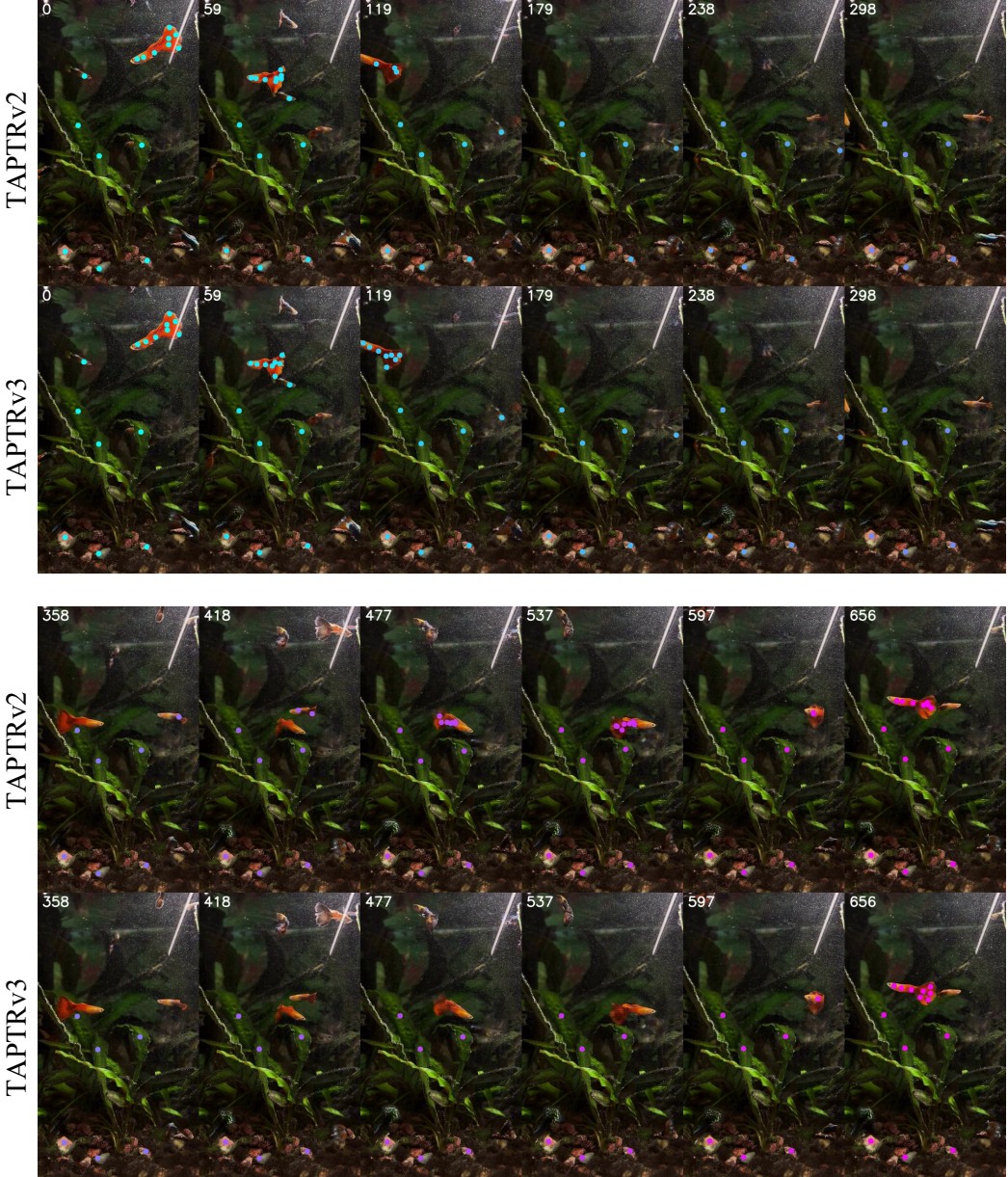

Figure 10: **Visual comparison between TAPTRv2 and TAPTRv3.** When the goldfish is about to swim out of the frame from right to left (119th frame), TAPTRv2 loses many target tracking points. Afterward, the goldfish swims back from left to right, and starting from the 358th frame, the video shows the other side of the goldfish, where the original target tracking points are occluded. However, TAPTRv2 incorrectly estimates them as visible or on another fish. TAPTRv3, on the other hand, maintains the correct estimation. Until the last dozens of frames, when the goldfish turns around again, TAPTRv3 successfully detects the initial target tracking points, estimates them as visible, and provides accurate positions. The corresponding videos (CompareVideo2_TAPTRv2.mp4 and CompareVideo2_TAPTRv3.mp4) are provided in the supplementary material.

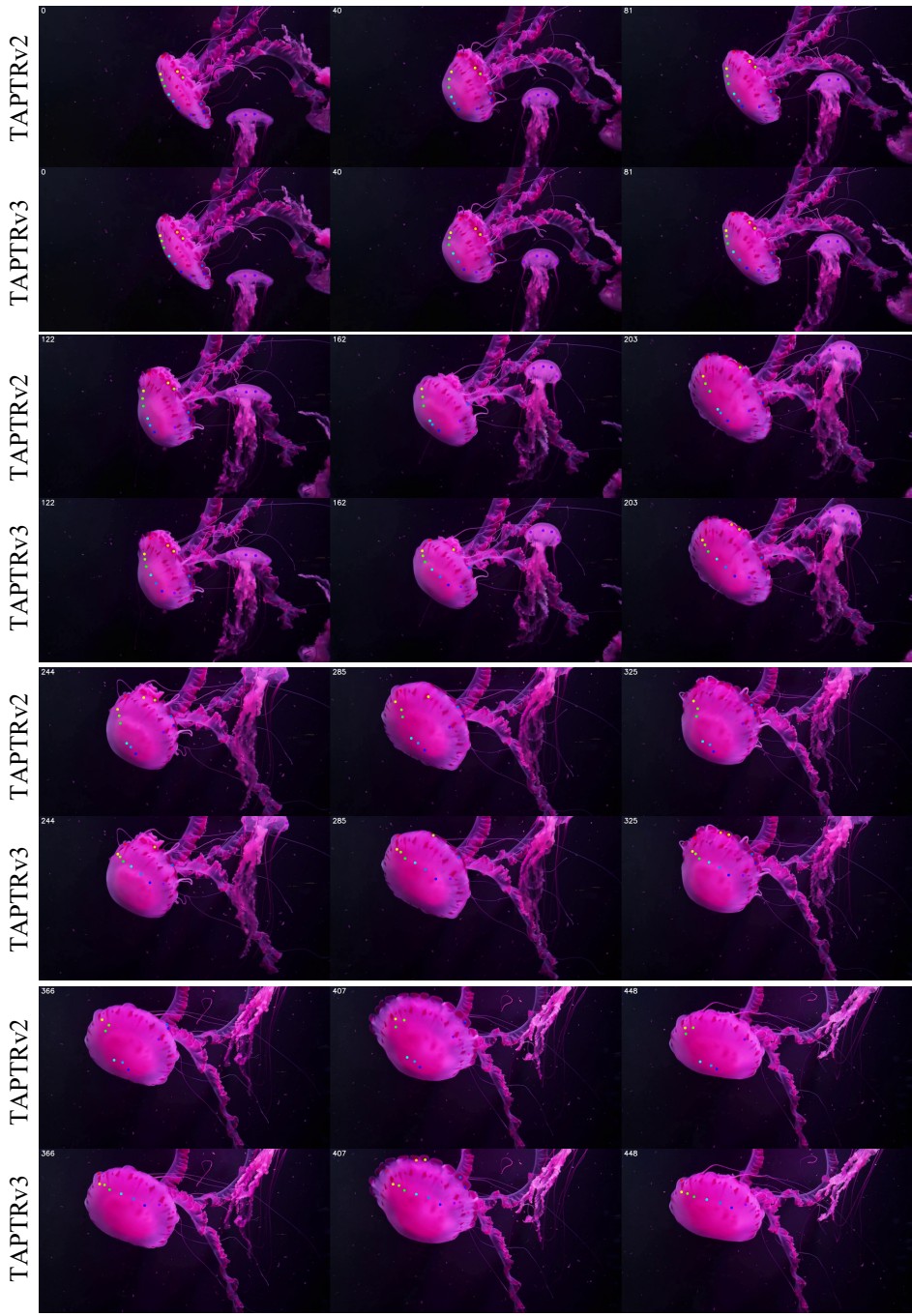

Figure 11: **Visual comparison between TAPTRv2 and TAPTRv3.** Best view in electronic version. Over time, TAPTRv2 incorrectly estimates the location and visibility of points on jellyfish, and the error accumulates, while TAPTRv3's results are more accurate. The corresponding videos (CompareVideo3_TAPTRv2.mp4 and CompareVideo3_TAPTRv3.mp4) are provided in the supplementary material.

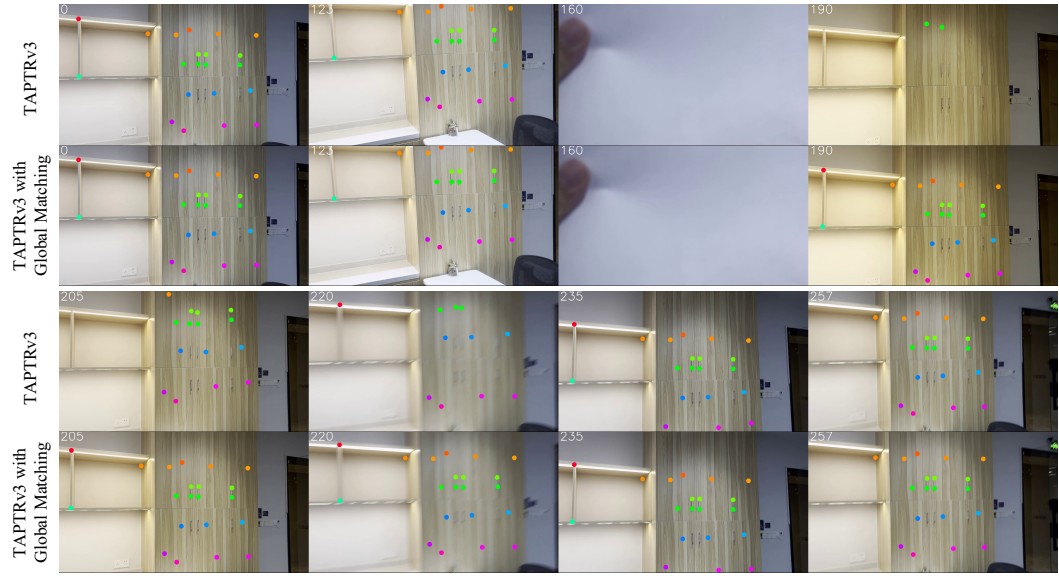

Figure 12: **Visual comparison between TAPTRv3 with and without the auto-triggered global matching.** After the occluder appears and then disappears, TAPTRv3 without auto-triggered global matching takes about 70 frames to successfully re-track the target tracking points. However, with the help of global matching, this process takes only two frames. The corresponding videos (CompareVideo4_TAPTRv3wGM.mp4 and CompareVideo4_TAPTRv3woGM.mp4) are provided in the supplementary material.

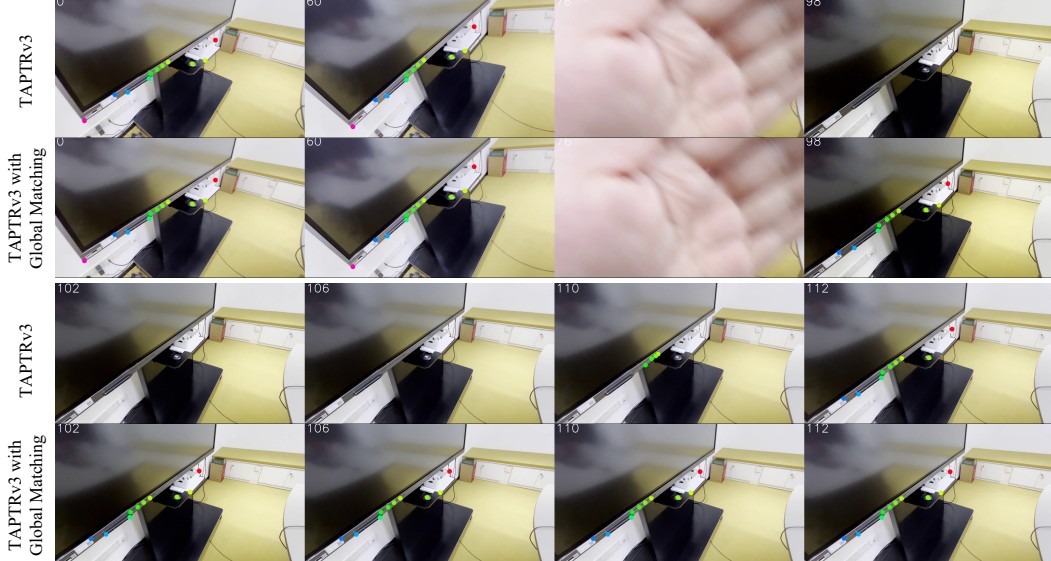

Figure 13: **Visual comparison between TAPTRv3 with and without the auto-triggered global matching.** After the occluder appears and then disappears, TAPTRv3 without auto-triggered global matching takes about 14 frames to successfully re-track the target tracking points. However, with the help of global matching, this process takes only two frames. The corresponding videos (CompareVideo5_TAPTRv3wGM.mp4 and CompareVideo5_TAPTRv3woGM.mp4) are provided in the supplementary material.

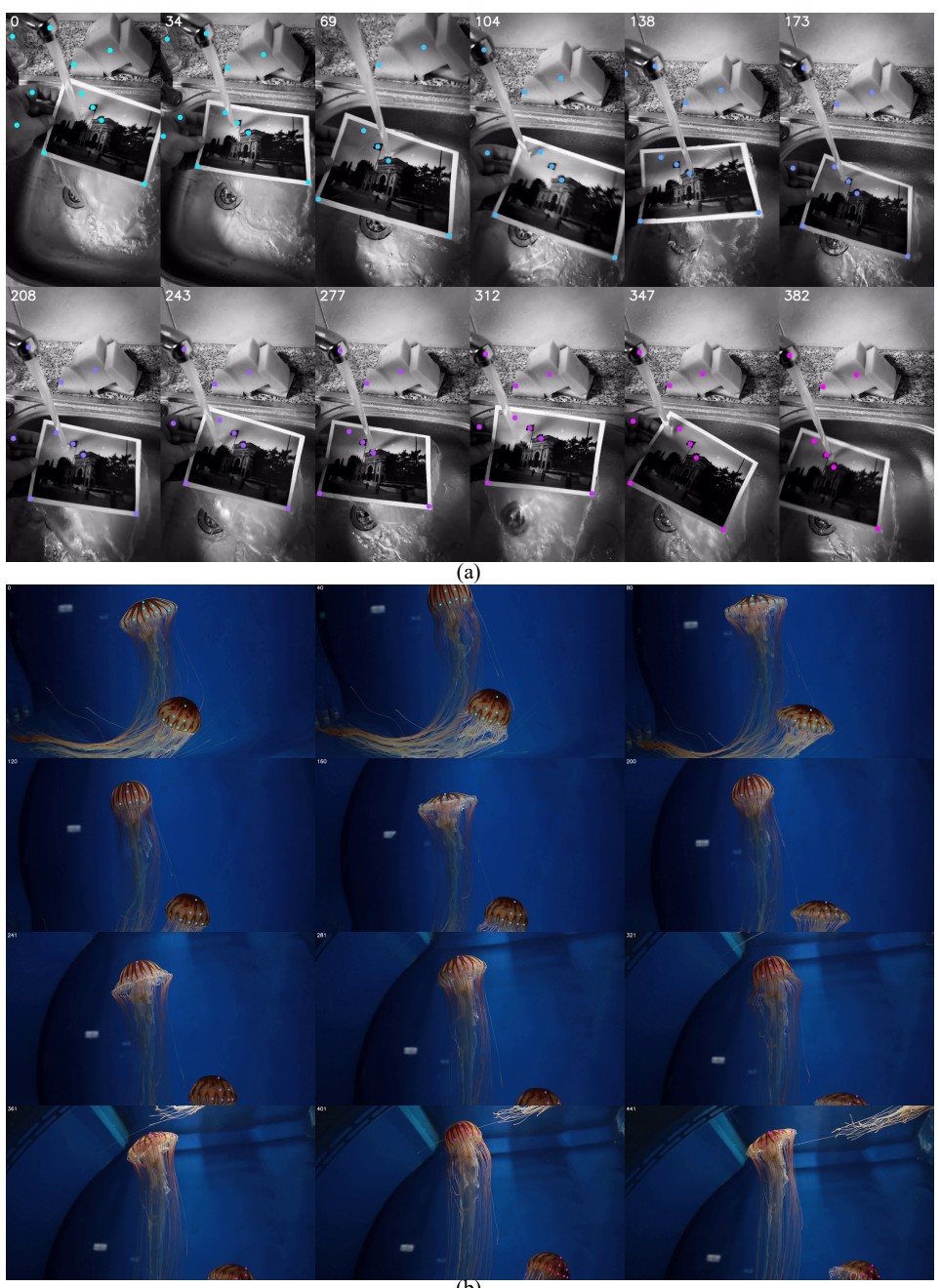

Figure 14: **Additional visualizations of TAPTRv3's robust predictions.** The corresponding videos (RobustVideo1.mp4 and RobustVideo2.mp4) are provided in the supplementary material.

