# OpenReview forum: "TAPTRv3: Spatial and Temporal Context Foster Robust Tracking of Any Point in Long Video"
_ICLR.cc/2026/Conference — ICLR 2026 Poster_

### Official Review · Reviewer_YtCz · 2025-10-28

**Soundness:** 4
**Presentation:** 3
**Contribution:** 3
**Rating:** 8
**Confidence:** 4

**Summary:**

This paper presents an enhanced version of TAPTRv2, a DETR-like framework for point tracking. The authors aim to mitigate the feature drifting issue that occurs in long video sequences by introducing several key modifications: (i) a patch-level similarity computation for refining attention weights, (ii) feature anchoring based on the initial frames, and (iii) aggregation of historical feature changes. Experimental results demonstrate consistent and substantial performance gains over the baseline.

**Strengths:**

The paper provides an updated and improved version of a well-recognized community benchmark in point tracking. The proposed enhancements, such as patch-level similarity computation in attention and initial feature anchoring, are methodically designed and empirically validated, leading to stable and meaningful improvements across benchmarks.

**Weaknesses:**

While the proposed techniques yield consistent quantitative improvements, the overall methodological novelty appears limited. Patch-level feature similarity computation has been explored in prior works such as Context-PIPs and Track-On. Similarly, leveraging initial-frame features and historical feature fusion are well-established practices in tracking literature, making it difficult to identify a distinct conceptual innovation in the paper.

Minor Issues:
- Figure 3 is not particularly illustrative. I recommend replacing it with a better qualitative example to better demonstrate the improvements.
- A visualization of point trajectories for multiple tracked points using a well-received benchmark video would be highly valuable for evaluating qualitative performance and motion consistency.

**Questions:**

Please see weakness

---

> ### Author Response · Authors · 2025-11-20
> **Response to Reviewer YtCz (Part 1/2)**
>
> We sincerely thank Reviewer YtCz for the strong acknowledgment of our method and extensive experimental results across benchmarks.
> Our detailed responses to your comments are as follows.
> The revised manuscript has been uploaded, with all changes clearly highlighted in blue for your convenience.
>
> ### [W1] The conceptual innovation is limited
>
> Thank you very much for this feedback.
> We would like to clarify that our innovation lies in carefully analyzing the limitations of existing DETR-like query-based frameworks for point tracking and addressing them effectively from both spatial and temporal perspectives.
> Below, we elaborate on the novelty of our approach and how it differs from prior work.
>
> 1. Spatial dimension
>
>     Our key insight is that TAPTRv2 [1] applies key-aware deformable attention [2] to explicitly update the features and positions of query points, but this mechanism was originally designed for object-level tasks.
>     In object detection and object tracking, each query encodes information about an entire object, making similarity computation robust and reliable. In contrast, point-level tasks cannot rely solely on the feature of a single point to accurately assess similarity for precise localization.
> Motivated by this, we innovatively propose to align the similarity computation in key-aware deformable attention with its object-level counterpart, which is the key insight behind the Context-aware Cross-Attention (CCA) module.
> The Contxet-PIPs [3] method you mentioned generates an extra correlation volume used as auxiliary information fed to an MLP-Mixer to aid trajectory refinement.
> Track-On [4] utilizes a coarse-to-fine approach, performing patch classification first, followed by further optimization.
> In contrast, our approach directly redesigns the attention weight calculation within TAPTRv2's core module to explicitly achieve more accurate similarity computation and position updates within a single frame.
> Therefore, we believe **both the motivation and implementation are distinct** from previous methods.
>
> 2. Temporal dimension
>
>     Our core idea is to treat the initial feature as an immutable identity anchor $f_0$ to prevent feature drift, while simultaneously using attention mechanism to aggregate historical information into a dynamic residual $\Delta f_t$ that adapts to appearance changes.
>     Building on this framework, we further innovate by introducing a visibility-aware weighting mechanism that uses the model's predicted visibility to weight historical frames, making $\Delta f_t$ more reliable.
>     This approach decouples the point's identity from its appearance changes, maintaining a balanced and robust formulation.
>     Although the utilization of initial features and historical information has been practiced in tracking, prior methods [4,5,6,7,8], to our knowledge, typically update the initial features or templates, which introduces the risk of feature drift.
>     In contrast, our method **remains faithful to the initial feature in both the spatial attention computation and the temporal modeling**.
>
> In summary, our contribution is not about isolated technical elements but about constructing a **conceptually coherent framework for point tracking through coordinated spatial and temporal innovations**.
> Both the **underlying insights and the specific implementations** differ from existing work.
> Ultimately, we successfully adapt the DETR-like architecture, originally designed for object-level tasks, to achieve highly competitive performance on point-level tracking.
> We believe our findings offer useful inspiration for future efforts toward unifying tracking across different levels of granularity.
>
> - [1] TAPTRv2: Attention-based Position Update Improves Tracking Any Point, NeurIPS 2024.
> - [2] Lite DETR : An Interleaved Multi-Scale Encoder for Efficient DETR, CVPR 2023.
> - [3] Context-PIPs: Persistent Independent Particles Demands Spatial Context Features, NeurIPS 2023.
> - [4] Track-On: Transformer-based Online Point Tracking with Memory, ICLR 2025.
> - [5] MixFormer: End-to-End Tracking with Iterative Mixed Attention, CVPR 2022.
> - [6] SeqTrack: Sequence to Sequence Learning for Visual Object Tracking, CVPR 2023.
> - [7] SUTrack: Towards Simple and Unified Single Object Tracking, AAAI 2025.
> - [8] CoTracker: It is Better to Track Together, ECCV 2024.

---

> > ### Author Response · Authors · 2025-11-20
> > **Response to Reviewer YtCz (Part 2/2)**
> >
> > ### [Minor Issue 1]
> >
> > Thank you very much for this valuable suggestion.
> > We agree that the original Figure 3 lacks sufficient visual impact. The tracked points in the video are sparse and not very prominent, which required readers to inspect the figure carefully and made it difficult to clearly demonstrate the improvements brought by our method.
> > Following your suggestion, we have **replaced it with a more illustrative visualization**. The revised figure is included in the updated manuscript, and we provide a brief description below.
> >
> > We selected a real-world video with 351 frames to showcase the improvement of our model.
> > The video captures a very long train passing through the scene.
> > At the beginning of the video, we generate query points using a $40 \times 40$ grid over the foreground region (excluding the sky). This example presents a typical long-term occlusion scenario of nearly 300 frames, posing a considerable challenge to model robustness.
> > As shown in the updated figure, TAPTRv2 makes large-scale visibility mispredictions once the train enters the frame, and after the train leaves, its predicted point locations become unstable, with most tracks lost.
> > In contrast, TAPTRv3 maintains stable and accurate tracking throughout the entire video, correctly predicts visibility during occlusion, and recovers accurate point locations once the train leaves the frame.
> > We believe this visualization is more intuitive than the previous one and more **clearly demonstrates the improvements** of our approach.
> >
> > We again sincerely thank you for this valuable suggestion, which helped us strengthen our work.
> > We also provide **additional real-world video visualizations in the Sec. A.8 of Appendix** to further illustrate the clear improvements of our method, such as in Figures 9 to 11.
> > Corresponding videos are also included in the supplementary material for more intuitive inspection.
> > We would be very happy to receive any further suggestions you may have.
> >
> > ### [Minor Issue 2]
> >
> > Thank you for this insightful suggestion and we fully agree with you.
> > Using videos from a well-received benchmark and visualizing the motion trajectories provides a more intuitive comparison and clearly illustrates the consistency of trajectory estimation.
> >
> > Following your suggestion, we adopted the visualization protocol used in CoTracker [8] and produced similar visualizations on the DAVIS dataset.
> > These results have been **added as Figure 8 in Appendix Sec. A.8**.
> > We believe this visualization offers a **more comprehensive qualitative demonstration** of our model's performance.
> > We welcome any further feedback you may have.
> >
> > ---
> >
> > We sincerely thank you again for the valuable comments provided during this review.

---

### Official Review · Reviewer_PNie · 2025-10-30

**Soundness:** 3
**Presentation:** 3
**Contribution:** 2
**Rating:** 6
**Confidence:** 4

**Summary:**

The authors build upon point tracking framework of TAPTRv2, where they propose an improved version called TAPTRv3 that achieves better performance on long-term scenarios. In the proposed tracking method, they introduce two main contributions where (1) context-aware cross-attention (CCA), and (2) visibility-aware long-temporal attention (VLTA) are the key components of this paper. To validate this, experimental evaluations including large-scale benchmarks of TAP-Vid and RoboTAP where the proposed algorithm shows improvements over the previous methods.

**Strengths:**

- The formulation for the proposed method is simple and reasonable, and contributions claimed by the authors are straightforward and clear.

- The motivation of the proposed method is clear, where failure cases of TAPTRv2 including scenarios with long-term temporal drift and scene cuts are well-addressed and quantified.

- The authors performed ablation experiments that validate the effectiveness of each proposed component. The results in Table 2 show that each proposed module (VLTA, CCA, etc.) adds to the gains in empirical performance.

**Weaknesses:**

- The main experimental comparisons are performed on the subsets of TAP-Vid benchmark, and albeit them being oriented for long-term tasks compared to previous benchmarks such as DAVIS, they seem still short in terms of temporal length. Datasets such as PointOddyssey [a] contain longer sequences and is more oriented for evaluating the proposed method.

  - [a] Yang et al., PointOdyssey: A Large-Scale Synthetic Dataset for Long-Term Point Tracking., ICCV 2023.

- Although the proposed framework somehow works by detecting scene cuts through the external library, it is not a trainable module with questionable generalizability issues. Since the main objective of the proposed framework is to deal with long-term videos, the overall framework should systematically include such algorithm in a integrated and optimized manner.

- Although the main objective of the proposed method is to tackle long-term scenarios well, Table 13 in the supplementary material shows slightly lower performance on short-term scenarios compared to other methods. Does this imply that the proposed CCA and VLTA modules hinder precise localization in shorter videos? Are there experimental analysis for the reason for the performance degradations?

**Questions:**

Please refer to the weaknesses section. Although the proposed scheme seems effective, limitations in the evaluations and performance degradations should be addressed.

---

> ### Author Response · Authors · 2025-11-20
> **Response to Reviewer PNie (Part 1/2)**
>
> We thank Reviewer PNie for acknowledging the sound rationale of our method, the improvement in experimental results, and the detailed ablation studies.
> Our detailed responses to your comments are as follows.
> The revised manuscript has been uploaded, with all changes clearly highlighted in blue for your convenience.
>
> ### [W1] Evaluation should be conducted on PointOdyssey with longer sequences
>
> Thank you for this valuable suggestion. We fully agree with your point.
> PointOdyssey [1] is a benchmark composed of synthetic data, and it is noticeably more realistic than TAP-Vid-Kubric. A key characteristic of PointOdyssey is its extremely long video sequences: the average length in the test set reaches 2386 frames, and the longest video contains 4325 frames, which far exceeds the video lengths in the TAP-Vid benchmark.
> Since our method is designed to address challenges in long videos, evaluation on PointOdyssey is **highly meaningful**.
>
> To this end, we directly evaluated TAPTRv3 trained only on the Kubric dataset on the PointOdyssey test set (including 12 valid videos).
> This allows us to assess the model's performance on longer and more challenging sequences, while also providing insight into its generalization ability.
> We report the metrics proposed in PointOdyssey, including $\delta_{avg}$, $\delta_{avg}^{vis}$, and $\delta_{avg}^{occ}$, which measure localization accuracy.
> The metrics $\delta_{avg}^{vis}$ and $\delta_{avg}^{occ}$ are computed in the same way as $\delta_{avg}$ but consider only visible and occluded points, respectively.
> In addition, we also report the Survival metric, which measures the average number of frames before tracking failure, where failure is defined as an error exceeding 50 pixels.
> The results are shown in the table below.
>
> |               | Traninig Data |$\delta_{avg}$ |$\delta_{avg}^{vis}$   |$\delta_{avg}^{occ}$   |Survival       |
> |:---           |:---:          |:---:          |:---:                  |:---:                  |:---:          |
> | TAP-Net [2]   |PointOdyssey   |28.4           |-                      |-                      |18.3           |
> | PIPs [3]      |PointOdyssey   |27.3           |-                      |-                      |42.3           |
> | PIPs++ [1]    |PointOdyssey   |29.0           |32.4                   |18.8                   |47.0           |
> | CoTracker [4] |PointOdyssey   |30.2           |32.7                   |24.2                   |55.2           |
> | Track-On [5]  |Kubric         |34.2           |38.1                   |-                      |49.5           |
> | TAPTRv2 [6]   |Kubric         |26.1           |28.4                   |21.5                   |50.0           |
> | TAPTRv3 (Ours)|Kubric         |33.9           |37.4                   |26.0                   |51.3           |
>
> Even without being trained on PointOdyssey, TAPTRv3 achieves **highly competitive performance**.
> More importantly, the improvement compared to TAPTRv2 is **extremely obvious**, especially a 9.0 point gain on the $\delta_{avg}^{vis}$ metric, which measures tracking accuracy on visible points.
> The experimental result validates that our insight into the limitations of TAPTRv2 in long videos is correct, and our proposed modules are **highly fruitful**.
>
> We are truly grateful for this insightful suggestion. It has helped make our evaluation more comprehensive, and we have **added a introduction of PointOdyssey and the corresponding evaluation results to Sec. A.3 of the Appendix**.
>
> - [1] PointOdyssey: A Large-Scale Synthetic Dataset for Long-Term Point Tracking, ICCV 2023.
> - [2] TAP-Vid: A Benchmark for Tracking Any Point in a Video, NeurIPS 2022.
> - [3] Particle Video Revisited: Tracking Through Occlusions Using Point Trajectories, ECCV 2022.
> - [4] CoTracker: It is Better to Track Together, ECCV 2024.
> - [5] Track-On: Transformer-based Online Point Tracking with Memory, ICLR 2025.
> - [6] TAPTRv2: Attention-based Position Update Improves Tracking Any Point, NeurIPS 2024.

---

> > ### Author Response · Authors · 2025-11-20
> > **Response to Reviewer PNie (Part 2/2)**
> >
> > ### [W2] The scene cut detector is not a trainable module
> >
> > Thank you for the insightful suggestion.
> > However, we would like to clarify that, due to **limitations in the available training data**, it is currently difficult for us to perform such joint training.
> > The Kubric dataset commonly used for this task contains synthetic data with a relatively singular domain.
> > Training a model solely on this dataset to explicitly or implicitly determine whether a frame contains a scene cut is highly challenging.
> > Therefore, we made a temporary compromise and adopted a widely recognized open-source method to handle scene cut detection separately. In practice, this traditional algorithm is **sufficiently accurate in most cases**.
> > As reported in Table 10 of the Appendix, on the Kinetics subset containing approximately 300 complex real-world videos, the auto-trigger global matching mechanism yields **a clear performance improvement**.
> >
> > We agree that your insight is highly profound. As you pointed out, our solution may still be imperfect. Nonetheless, it enables us to **achieve strong empirical results and validate our key intuition**: compared with relying on the model's prediction from the previous frame, the position estimated by global matching is less accurate, but its advantage lies in providing a fast, coarse global localization.
> > The two components should therefore be combined.
> > We believe that, with access to larger and more diverse training datasets in the future, it will become feasible to integrate scene cut detection into the overall framework for joint training, leading to improved performance and generalization.
> >
> > Once again, thank you for this valuable feedback. We have **added the corresponding explanation in the Appendix Sec. A.6**.
> >
> > ### [W3] Performance degradation on shorter videos
> >
> > Thank you for pointing out this issue.
> > We would like to clarify that the performance difference compared to our baseline, TAPTRv2, is only 0.3 AJ, which we consider a **reasonable variance within the experimental fluctuation range**.
> >
> > The DAVIS dataset contains only 30 relatively short video sequences. In comparison with other datasets, especially Kinetics which includes over 1100 videos, the performance on DAVIS exhibits much higher variance.
> > We further observed that the metric **could be further improved to match TAPTRv2** by simply modifying a few hyperparameters, such as the visibility classification threshold or the number of support points (a evaluation methodology proposed by CoTracker [4] and widely adopted by subsequent works like LocoTrack [7], Track-On [5], CoTracker3 [8], TAPNext [9], etc.).
> > We must emphasize that the experimental results reported in our paper are obtained using **the same set of hyperparameters across all datasets**.
> >
> > Finally, thank you once again for your feedback. As you noted, the primary objective of our work is to mitigate the feature drift issues in long videos.
> > When the video length is short, our two core modules are rarely activated, and the method essentially degenerates to TAPTRv2 case.
> > Therefore, achieving comparable performance to TAPTRv2 on the DAVIS dataset, which has a notably shorter average frames, is expected.
> > We believe this observation **does not affect our core contributions**.
> >
> > - [7] Local All-Pair Correspondence for Point Tracking, ECCV 2024.
> > - [8] CoTracker3: Simpler and Better Point Tracking by Pseudo-Labelling Real Videos, ICCV 2025.
> > - [9] TAPNext: Tracking Any Point (TAP) as Next Token Prediction, ICCV 2025.
> >
> > ---
> >
> > We sincerely thank you again for the valuable comments provided during this review.

---

### Official Review · Reviewer_c9Zu · 2025-11-01

**Soundness:** 3
**Presentation:** 2
**Contribution:** 2
**Rating:** 4
**Confidence:** 4

**Summary:**

This paper proposes TAPTRv2, a point tracking method. The core of this paper lies in presenting two corresponding operations: Context-aware Cross-Attention (CCA) and Visibility-aware Long-Temporal Attention (VLTA), which are designed to improve the quality of spatial cross-attention and the effectiveness of long-term feature updates. Meanwhile, to address scene switches, this paper proposes an auto-triggered global matching mechanism, which enhances tracking stability when scene switches occur.

**Strengths:**

- This paper focuses on improvements such as enhancing context-aware capabilities and modifying attention weights using visibility, which are highly intuitive and also suitable for video tasks.
- This paper achieves good performance on multiple datasets.
- The ablation experiments for the newly proposed modules in this paper are conducted in detail.

**Weaknesses:**

- The key improvement of this paper lies in proposing two types of attention mechanisms to replace previous RNN-like methods. However, there are now many improved RNN-like neural networks, such as readily applicable Mamba and RWKV. The paper lacks sufficient explanation for these methods; could improved recurrent structures also alleviate the challenges in long-context modeling?
- In visibility-aware attention, the authors modify attention weights using predicted visibility. While this operation is shown to be beneficial in results, another scenario is that errors in visibility prediction may in turn lead to performance degradation. The authors still need to explain why this operation can be guaranteed to have a positive effect.
- The implementation of CCA shares the same core idea as deformable attention.  And the paper in my opinion is more about combining and improving existing technologies.
- There are some ambiguities in the description of the method, such as the implementation of APU and the rationale behind such operations.
- The main reason for the acceleration in this paper is the use of a smaller backbone. However, could it be that the backbone inherently has little impact on this task? The authors still need to rule out this impact.

**Questions:**

- There are now many improved approaches for RNN-like neural networks, such as Mamba and RWKV. Why is it necessary to return to using attention? Is this truly a better choice compared to other structures?
- In visibility-aware attention, the authors modify attention weights using predicted visibility. While this operation is shown to be beneficial in results, are there cases where errors in predicting visibility lead to performance degradation instead? Why can it be guaranteed to have a positive effect?
- There are some ambiguities in the description of the method, such as the implementation of APU. What is the rationale behind such an operation? Is it merely due to the addition of parameters?
- Are point tracking methods sensitive to the backbone network?

---

> ### Author Response · Authors · 2025-11-20
> **Response to Reviewer c9Zu (Part 1/3)**
>
> We thank Reviewer c9Zu for acknowledging the sound rationale behind our new module design, the superiority of our experimental results, and the detailed ablation studies.
> Our detailed responses to your comments are as follows.
> The revised manuscript has been uploaded, with all changes clearly highlighted in blue for your convenience.
>
> ---
>
> ### [W1] & [Q1] Lack of explanation for using attention instead of Mamba or RWKV
>
> Thank you for this feedback.
> First of all, we would like to emphasize that our architectural improvements focus on two aspects: spatial modeling and temporal modeling. For the temporal dimension, we introduce the Visibility-aware Long-Temporal Attention (VLTA) module to replace the previous RNN-like design.
> Our choice to use attention instead of more advanced RNN-like structures such as Mamba [1] or RWKV [2] is based on the following considerations.
>
> We fully acknowledge that Mamba, RWKV, and other improved RNN-like architectures have demonstrated strong potential and efficiency.
> However, the core insight behind VLTA is that the **initial-frame information of the target point should be treated as the most reliable anchor and must be preserved throughout the tracking process**.
> Although Mamba and RWKV improve long-range information propagation through state compression and selective forgetting, they fundamentally **rely on a single evolving hidden state to carry information over time**. Regardless of how efficient the mechanism is, this hidden state represents a lossy compression of the entire history.
> In long videos, a point may disappear for hundreds of frames, during which the fine-grained appearance details stored in the hidden state may **degrade or become contaminated**.
>
> In contrast, our approach keeps the initial-frame anchor intact and uses attention to directly retrieve information from historical frames to accommodate appearance changes.
> Visibility prediction is further incorporated to ensure that more reliable historical information is aggregated.
>
> In summary, we believe that for video point tracking, hidden-state evolution is less suitable for handling the complex challenges of long videos.
> While attention may incur higher computational cost, by appropriately controlling the size of the historical buffer, our model still **achieves a good balance between performance and efficiency**.
> As described in Appendix Sec. A.2, TAPTRv3 can process arbitrarily long videos in a streaming manner (tracking 100 points simultaneously with less than 2GB GPU memory), making it **deployment cost-efficient**.
>
> Once again, thank you for this insightful feedback. We have **added the corresponding explanation in the Appendix Sec. A.6** to address potential concerns.
>
> - [1] Mamba: Linear-Time Sequence Modeling with Selective State Spaces, COLM 2024.
> - [2] RWKV: Reinventing RNNs for the Transformer Era, EMNLP 2023.
>
> ### [W2] & [Q2] Further explanation of the effectiveness of visibility-aware attention
>
> Thank you for raising this question.
> First, we apply the predicted visibility in a **soft weighting manner**, where frames predicted as more likely to be invisible are down-weighted according to confidence rather than discarded.
> The operation is intuitively sound when the visibility prediction is correct. When the visibility estimation is erroneous, it can be further categorized into the following two specific cases:
>
> 1. Visible point -> predicted as invisible
>
>     In this case, the attention weights for these frames are reduced, leading the model to extract less historical information from them.
>     As a result, the model relies more on the earlier frames where the point is visible, and these frames provide reliable information.
>
> 2. Invisible point -> predicted as visible
>
>     This case degenerates to the situation where visibility weighting is not applied.
>     The model may aggregate unreliable historical information, similar to the baseline without visibility-aware weighting.
>
> In summary, the visibility-aware attention mechanism **provides an overall positive effect**.
> Thanks for your review, and we have **added the corresponding explanation in the Appendix Sec. A.6** to address potential concerns.

---

> > ### Author Response · Authors · 2025-11-20
> > **Response to Reviewer c9Zu (Part 2/3)**
> >
> > ### [W3] Similarity between CCA and deformable attention
> >
> > Thank you for raising this concern. First, we would like to clarify that the proposed Context-aware Cross-Attention (CCA) module is an **extension of the key-aware deformable attention [3] mechanism for point-level tasks**, designed to achieve more accurate and robust feature querying and position updates.
> >
> > TAPTRv2 [4] uses key-aware deformable attention to explicitly compute the similarity between a query point and its sampled neighbors, and then updates features and positions based on the resulting attention weights.
> > However, this operation was **originally designed for object-level tasks**, where each query feature represents an entire object, making the mechanism effective.
> > The point tracking task requires more fine-grained spatial perception, and the point-level query and sampled keys are highly local and therefore more susceptible to noise.
> > Based on this insight, we modify this operation for point-level tracking by retaining the query point's original patch-level information and using it to query richer spatial context blocks.
> > This **compensates for the gap between point-level and object-level representations**.
> > The results in Tables 2 and 3 strongly support the effectiveness of this design.
> >
> > In summary, we believe the proposed CCA module is not merely a simple concatenation of existing operations.
> > The core idea of the deformable attention [5] you mentioned is to achieve higher efficiency and faster convergence through sparse computation.
> > One of our core contributions is **successfully adapting this efficient mechanism to point tracking while maintaining superior performance**.
> >
> > - [3] Lite DETR : An Interleaved Multi-Scale Encoder for Efficient DETR, CVPR 2023.
> > - [4] TAPTRv2: Attention-based Position Update Improves Tracking Any Point, NeurIPS 2024.
> > - [5] Deformable DETR: Deformable Transformers for End-to-End Object Detection, ICLR 2021.
> >
> > ### [W4] & [Q3] Ambiguities in the description of the implementation of APU
> >
> > Thank you for your feedback.
> > In fact, the Attention-based Position Update (APU) is a method **originally proposed in TAPTRv2, where its rationale and effectiveness have already been validated**.
> > For clarity, we provide a brief explanation here.
> >
> > The core idea of APU is to directly and explicitly calculate the similarity between the query point and surrounding sampled points to weight the relative positional offsets, thereby completing a position update, rather than solely relying on input features to update the position via regression, as done previously.
> > Conceptually, this process can be viewed as a special form of attention operation, where the values are not image features but positional vectors.
> > This explicit similarity-based position update approach inherently has a clear technical rationale.
> > Furthermore, it decouples the query point's content feature update from the position update, which to a certain extent makes the query point's features less prone to contamination.
> >
> > In TAPTRv3, our improvements primarily focus on refining the similarity computation in APU through the proposed CCA module, making it better suited for point-level tasks. Please refer to [W3] for details.

---

> > > ### Author Response · Authors · 2025-11-20
> > > **Response to Reviewer c9Zu (Part 3/3)**
> > >
> > > ### [W5] & [Q4] Need to evaluate the impact of backbone on the model
> > >
> > > Thanks for this valuable suggestion.
> > > We indeed observed in our experiments that replacing the backbone of TAPTRv3 from ResNet50 with the more lightweight ResNet18 yielded comparable performance.
> > > To address your confusion, we have conducted **further experiments** on the TAP-Vid benchmark [6], reporting the Average Jaccard (AJ) metric.
> > >
> > > |Model          |Backbone               |DAVIS  |Kinetics   |RGB-Stacking   |
> > > |:---           |:---                   |:---:  |:---:      |:---:          |
> > > |TAPTRv3(Ours)  |ResNet18               |63.2   |54.9       |72.3           |
> > > |TAPTRv3(Ours)  |ResNet50               |63.2   |54.5       |73.0           |
> > > |TAPTRv3(Ours)  |Vit-Adapter with DINOv2|62.2   |54.2       |67.8           |
> > >
> > > We employ a larger-capacity backbone, ViT-Adapter [7] with DINOv2, and report the results in the table above. Overall, the performance decreases on all datasets, with a particularly notable drop on the synthetic RGB-Stacking dataset.
> > > However, this observation may not generalize to all point tracking methods. For example, to the best of our knowledge, Track-On [8] achieves a clear performance gain when replacing the ResNet backbone with ViT-Adapter.
> > > We believe that the discrepancy may **relate to differences in the overall model architecture**.
> > > For instance, our approach adopts a DETR-like design that is distinct from other point tracking frameworks.
> > > Our model includes encoder layers in addition to the backbone to further aggregate features.
> > > To better understand this behavior in our model, we additionally conduct experiments examining the impact of encoder layers, and summarize the results in the table below.
> > >
> > > |Model          |Backbone               |Encoder Layers |DAVIS  |Kinetics   |RGB-Stacking   |
> > > |:---           |:---                   |:---:          |:---:  |:---:      |:---:          |
> > > |TAPTRv3(Ours)  |ResNet18               |1              |61.6   |53.5       |66.7           |
> > > |TAPTRv3(Ours)  |ResNet18               |2              |63.2   |54.9       |73.0           |
> > > |TAPTRv3(Ours)  |ResNet18               |3              |58.7   |51.9       |62.6           |
> > >
> > > When using two encoder layers, our model achieves the best performance, and the impact of encoder depth is substantially larger than that of the backbone.
> > > We believe that using only a single encoder layer may lead to insufficient feature extraction, making the model capacity too limited relative to the task difficulty.
> > > In contrast, using three encoder layers noticeably slows convergence and increases training difficulty.
> > > It is worth noting that replacing the backbone or adjusting encoder depth typically **requires extensive hyperparameter tuning** (e.g., learning rate, weight decay) to achieve optimal performance.
> > > Constrained by time and computational resources, we leave this for future exploration.
> > >
> > > Furthermore, we reduce the model size of TAPTRv2 by replacing the ResNet50 backbone with ResNet18, following the design choice in TAPTRv3.
> > > As shown in the table below, the model exhibits a **significant performance drop** with the lighter backbone, which to some extent **reflects the effectiveness of our newly introduced components in the decoder**.
> > >
> > > |Model          |Backbone               |DAVIS  |Kinetics   |RGB-Stacking   |
> > > |:---           |:---                   |:---:  |:---:      |:---:          |
> > > |TAPTRv2        |ResNet18               |60.0   |45.6       |46.7           |
> > > |TAPTRv2        |ResNet50               |63.5   |49.7       |53.4           |
> > >
> > > Finally, we would like to clarify that another reason for adopting ResNet18 in our model is to enable a **fair comparison** with other excellent works in the community, such as TAP-Net [6], TAPIR [9], and CoTracker [10], as they mostly utilize backbones with a similar magnitude of parameters.
> > >
> > > Thank you once again for the suggestion regarding the supplementary experiments to further investigate the influence of the backbone on the model.
> > > We have **added the above experimental results to Section A.4 of the Appendix**, providing greater detail.
> > >
> > > - [6] TAP-Vid: A Benchmark for Tracking Any Point in a Video, NeurIPS 2022.
> > > - [7] Vision Transformer Adapter for Dense Predictions, ICLR 2023.
> > > - [8] Track-On: Transformer-based Online Point Tracking with Memory, ICLR 2025.
> > > - [9] TAPIR: Tracking Any Point with per-frame Initialization and temporal Refinement, ICCV 2023.
> > > - [10] CoTracker: It is Better to Track Together, ECCV 2024.
> > >
> > > ---
> > >
> > > We sincerely thank you again for the valuable comments provided during this review.

---

### Official Review · Reviewer_E5oP · 2025-11-01

**Soundness:** 2
**Presentation:** 3
**Contribution:** 2
**Rating:** 4
**Confidence:** 4

**Summary:**

This work presents TAPTRv3, which aims to leverage both spatial and temporal context to for robust tracking in long videos. It introduces two key operations, i.e. Context-aware Cross- Attention and Visibility-aware Long-Temporal Attention, which can improve the quality of spatial cross attention and long-term feature updating. An auto-triggered global matching mechanism is further triggered when a scene cut is detected. The tracker has been evaluated on public benchmarks.

**Strengths:**

1. The introduction of Context-aware Cross-Attention (CCA) and Visibility-aware Long-Temporal Attention (VLTA) seems reasonable.
2. The auto-triggered global matching mechanism is easy to follow.
3. This paper provides extensive quantitative results on the public tracking benchmarks, demonstrating the method's robustness, and efficiency.

**Weaknesses:**

1. The overall framework of TAPTRv3 is highly similar to TAPTRv2. In TAPTRv2, it also introduces Attention-based Position Update and visibility classifier to maintain the temporal consistency. Despite the implementation variations, the key idea of this work is not very novel.
2. The performance gains compared to CoTracker3 and Track-On in Table 1 are marginal.
3. The auto-triggered global matching is simply a global redetection mechanism, which has been widely explored in other tracking frameworks.

**Questions:**

1. What are the major insights of this work compared to TAPTRv2?
2. What’s the key difference between the proposed auto-triggered global matching and other redetection approaches?

---

> ### Author Response · Authors · 2025-11-20
> **Response to Reviewer E5oP (Part 1/3)**
>
> We thank reviewer E5oP for acknowledging the sound rationale behind our new module design and the positive experimental results.
> Our detailed responses to your comments are as follows.
> The revised manuscript has been uploaded, with all changes clearly highlighted in blue for your convenience.
>
> ---
>
> ### [W1] & [Q1] TAPTRv3 is highly similar to TAPTRv2, what are the major insights
>
> We appreciate your feedback regarding the potential similarities between our model and TAPTRv2 [1].
> However, we would like to emphasize that our main insight and contribution lie in identifying and addressing the **limitations in both spatial and temporal modeling** of TAPTRv2, which lead to severe feature drift over time in long video scenarios.
> TAPTRv2 formulates point tracking as an extension of object detection and adopts a DETR-like architecture.
> However, simply migrating a single-frame, object-level detection model to a point-level tracking task with strong temporal dependencies presents several challenges that must be addressed.
> We elaborate on our key insights from the perspective of both the spatial dimension and the temporal dimension below.
>
> 1. Spatial dimension
>
>     In the spatial dimension, TAPTRv2 introduces the Attention-based Position Update (APU) operation, which explicitly uses key-aware deformable attention to compute similarities between the query point and sampled local points, and then updates the query position by weighting their relative offsets.
>     However, TAPTRv2 simply uses features obtained via bilinear interpolation to compute similarity. This approach neglects the significant difference between such features and object-level features, making the process sensitive to noise and unreliable, especially in long videos where changes are more complex.
>
>     To address this issue, we propose the Context-aware CrossAttention (CCA) operation, which **extends the key-aware deformable attention mechanism to point-level tasks**. CCA computes patch-level similarity and leverages richer spatial context, effectively mitigating the limitations of the original APU computation. As a result, position updates become more accurate and robust.
>
> 2. Temporal dimension
>
>     In the temporal dimension, TAPTRv2 adopts a sliding-window strategy and initializes each window using the final frame's prediction from the previous window.
>     This RNN-like approach accumulates errors over long videos, ultimately causing the tracked point's features to drift or even vanish.
>
>     In TAPTRv3, we abandon the sliding-window update and introduce Visibility-aware Long-Temporal Attention (VLTA) module.
>     Our core insight is to **treat the initial frame's point information as the most reliable anchor** and maintain it throughout tracking.
>     To handle appearance changes over time, this anchor actively attends to historical frames.
>     Furthermore, we incorporate **predicted visibility to weight the attention**, ensuring that features are not aggregated from frames or regions where the point is not visible, thus improving long-term reliability.
>
>     Regarding the visibility classifier you mentioned, predicting point visibility is a fundamental requirement of the Tracking Any Point task. Consequently, **nearly all existing methods adopt an MLP as the visibility classifier**.
>
> We thank you again for raising this question. The concern may stem from the fact that we retain the DETR-like framework of TAPTRv2.
> However, as outlined above, we have conducted a thorough analysis of TAPTRv2's design limitations and **introduce substantial improvements** in both spatial and temporal modeling.
> The **significant performance gains over TAPTRv2** on benchmark datasets (Table 1) and the **detailed ablation studies** (Tables 2-6) strongly support our insights.
>
> - [1] TAPTRv2: Attention-based Position Update Improves Tracking Any Point, NeurIPS 2024.

---

> > ### Author Response · Authors · 2025-11-20
> > **Response to Reviewer E5oP (Part 2/3)**
> >
> > ### [W2] Comparison with prior methods
> >
> > Thank you for your comment. First of all, it is important to emphasize that, as shown in Table 1, our model **outperforms previous methods across the majority of datasets and metrics**. In particular, our model **demonstrates comprehensive superiority on the Average Jaccard (AJ) metric**, which considers the precision of both position and visibility predictions.
> > The CoTracker3 [2] and Track-On [3] you mentioned are both highly commendable works within the community, and we provide a more detailed comparison beyond just performance metrics below.
> >
> > 1. Comparison with CoTracker3: CoTracker3 is a well-recognized work. We believe its core contribution lies in proposing an exceptionally efficient semi-supervised training paradigm. By using pseudo-labels on real-world videos, it significantly boosts tracking performance.
> > In contrast, our core contribution is an **architectural innovation** that addresses the inherent limitations of DETR-like tracking models in long videos, resulting in more accurate and robust spatial and temporal feature querying.
> > When both models are trained solely on synthetic data, our model maintains clear superiority.
> > Furthermore, CoTracker3's online model still relies on a windowed input format, while our model accepts frame-by-frame input, enabling streaming processing.
> >     Finally, we believe our contributions are **highly complementary**, and combining our model design with CoTracker3's data and training strategy represents an exciting direction for future research.
> >
> > 2. Comparison with Track-On: Track-On is also an excellent work, dedicated to solving the long-term point tracking problem in a frame-by-frame online manner. It incorporates spatial and context memory to allow query points to adapt to appearance changes, essentially relying on an incremental strategy.
> > However, our core insight is different. **We treat the initial features as the most reliable anchor point**, ensuring they are not easily contaminated or updated. Simultaneously, to adapt to appearance variations, we query historical frames based on visibility and dynamically aggregate the historical information.
> > Furthermore, our model is more lightweight. While Track-On uses ViT-Adapter [4] with DINOv2, we only utilize ResNet18 as our backbone. This results in a **smaller parameter count (24.5 M vs. 49.1 M) and faster inference speed (29.8 FPS vs. 23.0 FPS)**. The inference speed was tested on the DAVIS dataset using the same RTX 3090 GPU with frame-by-frame input. Finally, Track-On requires different hyperparameter settings to achieve optimal performance across various datasets, whereas our model has no extra hyperparameter tuning process.
> >
> > - [2] CoTracker3: Simpler and Better Point Tracking by Pseudo-Labelling Real Videos, ICCV 2025.
> > - [3] Track-On: Transformer-based Online Point Tracking with Memory, ICLR 2025.
> > - [4] Vision Transformer Adapter for Dense Predictions, ICLR 2023.

---

> > > ### Author Response · Authors · 2025-11-20
> > > **Response to Reviewer E5oP (Part 3/3)**
> > >
> > > ### [W3] & [Q2] Key difference between our auto-triggered global matching and other redetection approaches
> > >
> > > Thank you for raising this concern. We must clarify that we do not claim the global matching operation itself as our novelty.
> > > Instead, **we propose the novel auto-trigger mechanism built on top of it as part of our core contribution.**
> > > As stated in our Appendix Sec. A.1:
> > > > "To avoid misunderstanding, we emphasize that our main contribution in global matching lies in the novel auto-trigger mechanism, rather than the global matching itself."
> > >
> > > Regarding this point, our core insight can be summarized as follows: although the locations provided by global matching are less accurate than the model's predictions from the previous frame, its key strength is the ability to quickly provide a coarse global position. Therefore, **it is crucial to leverage the strengths of both**.
> > >
> > > More specifically, as demonstrated by the experimental results in Table 5 of our main paper, relying solely on global matching for per-frame updates performs worse than using the predictions from the previous frame.
> > > However, we also observe that, due to biases in the training data, the magnitude of positional updates between adjacent frames is generally not too drastic.
> > > Our DETR-like model tends to search locally around the current position for feature matching.
> > > Consequently, during inference, when a shot change occurs (such as a scene cut), the model may require several frames to re-locate the target points (not necessarily causing complete tracking failure).
> > > Introducing global matching precisely in these cases can effectively mitigate this issue, **as illustrated in Figures 12 and 13 in the Appendix.**
> > > Leveraging the insight discussed above, **our model demonstrates superior performance in conventional video scenarios and is capable of swiftly recovering tracking after a scene cut.**
> > >
> > > We thank you for your valuable feedback. To reduce potential confusion, **we have added the clarification to Sec. 3.4 of the main text as well**.
> > >
> > > ---
> > >
> > > We sincerely thank you again for the valuable comments provided during this review.

---

### Author Response · Authors · 2025-12-01
**Summary of Author Response**

**Dear Reviewers, Area Chairs, and Program Chairs,**

We understand the recent incident on OpenReview has created unprecedented challenges for the reviewing process and we are profoundly grateful for the additional effort the Area Chairs have devoted under such circumstances.
Since none of the reviewers had responded to our rebuttal before the incident occurred, we have summarized the original reviews and our responses below, hoping to alleviate the burden on the ACs.

---

First, we thank all reviewers for their affirmation of our work.
**All four reviewers recognized the soundness of our motivation and methodology, as well as the strong performance of our model across multiple benchmarks.**

Reviewer E5oP noted that our auto-triggered global matching mechanism is easy to follow.
Reviewers c9Zu and PNie praised the thorough ablation studies.
Reviewer YtCz provided an overall positive evaluation of the framework and our contributions.

---

In addition, we have **thoroughly addressed all questions and concerns raised by the reviewers** and revised the manuscript accordingly (highlighted in blue).
A summary of our responses is provided below.

- **Elaboration**
  - As suggested by Reviewer E5oP, we clarified the fundamental advancements over TAPTRv2 and explained how the proposed auto-triggered global matching mechanism differs from redetection approaches used in other methods.
  - As suggested by Reviewer YtCz, we clarified our technical and conceptual novelty in greater detail and compared them with related works both within and outside this field.
- **Technical Details**
  - As suggested by Reviewer c9Zu, we provided additional explanations on why we adopt attention instead of Mamba or RWKV, analyzed the empirical effectiveness of the proposed visibility-aware attention, clarified the differences between our CCA and deformable attention, and introduced the APU module proposed in TAPTRv2 to resolve the ambiguities.
  - As suggested by Reviewer PNie, we explained the rationale behind not designing the scene cut detector as a trainable module
- **Experiments**
  - As suggested by Reviewer E5oP, we further emphasized performance gains over prior works and provided additional comparisons, including training data, model size, and inference speed.
  - As suggested by Reviewer c9Zu, we added new experiments investigating the influence of the backbone and the number of encoder layers.
  - As suggested by Reviewer PNie, we conducted a more comprehensive evaluation on the PointOdyssey benchmark and confirmed no performance degradation on short videos.
- **Visualizations**
  - As suggested by Reviewer YtCz, we replaced the visualization in Figure 3 with a more illustrative example and added new visualizations on well-received benchmark videos to show the consistency of trajectory estimation.

---

Best regards,

The Authors of Submission 7206

---

### Meta-Review · Area_Chair_LfMi · 2026-01-05

**Summary:**

The paper presents TAPTRv3 for online long-video point tracking, targeting long-term drift and abrupt discontinuities. It uses visibility-aware long-temporal attention to leverage long-range history under occlusion, context-aware cross-attention to reduce local matching ambiguity, and auto-triggered global matching to re-anchor tracking at scene cuts. It also removes the sliding-window dependency to better support streaming inference, and reports consistent improvements with extensive ablations and competitive results across benchmarks.

Reviewers focused on whether the contribution is substantially novel or mainly an incremental refinement, whether the experimental evidence convincingly supports long-video robustness, and whether scene-cut handling is reliable given its reliance on an external detector. Some discussion also considered whether improvements might be influenced by implementation choices such as backbone or inference settings, and whether there are trade-offs in short-video regimes.

**Reviewer Concerns:**

**Long-sequence robustness and evaluation coverage.**
Reviewer PNie questioned whether the original evidence was sufficient to support robustness on truly long videos and requested stronger long-sequence evaluation. The rebuttal responded by adding longer-horizon evaluations and additional comparisons and analyses, which improves confidence that the gains persist on long sequences and clarifies where the method is most effective.

**Novelty and differentiation from prior work.**
Reviewer E5oP and Reviewer YtCz raised concerns that the pipeline resembles established tracking patterns and is close to prior versions, making the conceptual contribution appear incremental. The rebuttal clarified the motivations and roles of the proposed modules and tightened the positioning, but the discussion still treats the contribution as an incremental refinement rather than a fundamentally new approach.

**Scene-cut handling and external dependency.**
Reviewer PNie flagged the reliance on an external scene-cut detector to trigger global matching as a potential generalization risk. The rebuttal clarified how the trigger and re-anchoring operate and acknowledged the design choice as pragmatic, but it does not remove the dependency or provide an integrated alternative, so the concern remains partly outstanding.

**Potential confounds from implementation choices.**
Reviewer c9Zu questioned whether accuracy and efficiency gains could be driven by backbone choice or inference settings rather than the proposed modules. The rebuttal added analyses and supporting evidence that reduce this concern and make the improvements more credible.

**Method clarity and mechanistic details.**
Reviewer c9Zu raised requests for clearer explanations of how visibility signals are used and how local matching is computed. The rebuttal provided clarifications that reduce ambiguity and make the pipeline easier to interpret.

**Reviewer Scores:**

Reviewer YtCz remained strongly supportive, emphasizing the strength and clarity of the empirical results and the consistency of the ablation evidence. Reviewer PNie focused on whether long-video robustness was sufficiently demonstrated and whether the scene-cut mechanism would generalize. The rebuttal’s added long-sequence results and clarifications could plausibly increase their confidence, although there is no explicit indication that they would formally raise their score. Reviewer E5oP and Reviewer c9Zu remained cautious, primarily due to concerns about incremental novelty and the external scene-cut dependency. While the rebuttal improved clarity and evidence, these points are not fully resolved.

Overall, despite improved clarity after rebuttal, the balance between strengths and weaknesses remains largely unchanged.

---

### Decision · Program_Chairs · 2026-01-26

Accept (Poster)